# Photonics-integrated terahertz transmission lines

Yazan Lampert [1,2,5] ✉, Amirhassan Shams-Ansari[3,4,5] ✉, Aleksei Gaier[1,2], Alessandro Tomasino[1,2], Xuhui Cao[1,2], Leticia Magalhaes[3], Shima Rajabali[1,2,3], Marko Lončar [3] & Ileana-Cristina Benea-Chelmus [1,2] ✉

Modern communication and sensing technologies connect the optical domain with the microwave domain. Accessing the terahertz region from 100 GHz to 10 THz is critical for providing larger bandwidths capabilities. Despite progress in integrated electronics, they lack a direct link to the optical domain, and face challenges with increasing frequencies ( > 1 THz). Electro-optic effects offer promising capabilities but are currently limited to bulk nonlinear crystals, missing out miniaturization, or to sub-terahertz bandwidths. Here, we address these challenges by realizing photonic circuits that integrate terahertz transmission lines on thin-film lithium niobate (TFLN). By providing terahertz field confinement and phase-matched interaction with optical fields, our miniaturized devices support low-noise and broad bandwidth terahertz generation and detection spanning four octaves (200 GHz to > 3 THz). By leveraging photonics' advantages in low-loss and high-speed control, our platform achieves control over the terahertz spectrum and its amplitude, paving the way for compact and power-efficient devices with applications in telecommunications, spectroscopy, quantum electrodynamics and computing.

Terahertz technologies, with operating frequencies between 0.1–10 THz, hold transformative potential across a broad range of applications, including high-speed communication (6G technology)[1–3], non-destructive imaging[4–7] and sensitive spectroscopy[8,9]. However, despite tremendous progress made, high-frequency electronic sources and detectors[10], such as multiplier chains[11] or plasma discharge[12] lose efficiency with increasing frequencies, accentuating the need for efficient detectors and emitters. Given these tremendous challenges, one attractive approach is to detect and generate these high frequencies through non-electronic techniques such as nonlinear optical mixing. Two distinct strategies have emerged. One approach involves $\chi^{(2)}$ materials that enable terahertz generation and detection through frequency mixing of optical and terahertz signals. In this case, the energy of the optical photons is lower than the bandgap of the $\chi^{(2)}$ medium. A second approach involves photomixing inside semiconductors,

requiring the optical photons to be absorbed above the bandgap[13]. In recent years, integrated photonics has experienced an accelerated growth with developments spanning both material categories, from lithium niobate (LN)[14], lithium tantalate (LT)[15], barium titanate (BTO)[16], to III-V on insulator, including gallium phosphide (GaP)[17], gallium arsenide (GaAs)[18] or indium phosphide (InP)[19]. The integration of terahertz systems within waveguide-based photonic structures is particularly appealing[20] because they offer smaller size, weight, and power (SWaP). In the context of sensing and spectroscopy, they offer confinement of terahertz radiation below the diffraction limit, increasing the spatial overlap with the sensed medium, boosting sensitivity[21,22]. By combining these benefits with other engineering knobs, that are enabled through waveguide engineering, nonlinear frequency mixing within photonic integrated circuits has the potential to provide a flexible and power-efficient method for generating and detecting terahertz radiation[23].

[1]Hybrid Photonics Laboratory, École Polytechnique Fédérale de Lausanne (EPFL), Lausanne, Switzerland. [2]Center for Quantum Science and Engineering (EPFL), Lausanne, Switzerland. [3]Harvard John A. Paulson School of Engineering and Applied Sciences, Harvard University, Cambridge, MA, USA. [4]DRS Daylight Solutions, 16465 Via Esprillo, San Diego, CA, USA. [5]These authors contributed equally: Yazan Lampert, Amirhassan Shams-Ansari. ✉e-mail: yazan.lampert@epfl.ch; amirhassan.shams-ansari@drs.com; cristina.benea@epfl.ch

Recent advances focus particularly on III-V platforms which integrate lasers[24–26], modulators[27] voltage-controlled amplifiers[28] and photodiodes with the photonic circuit. On-chip continuous-wave applications have shown great promise, such as generation up to 4 THz on InP substrates[29] or fully integrated generation and detection at 100-500 GHz using GaAs/AlGaAs quantum wells[30]. Recently, the integration of optical waveguides with photoconductive receivers has led to 22-fold increase in response, 500-fold improvement in terahertz responsivity, and a 4.7-fold reduction in noise-equivalent power compared to top illuminated PCAs[31]. Nevertheless, the efficiency of continuous-wave devices strongly decreases with frequency, with a typical -20 dB power roll-off around 0.5–1 THz[29], resulting in non-flat spectra. While continuous-wave terahertz applications have benefited from photonic integration, pulsed terahertz applications in these platforms are challenging, since they would require guiding femtosecond pulses for optical rectification and electro-optic sampling. Since this needs to be accomplished without significant loss, passive waveguides need to be realized from a different material, such as InP[31,32], GaAs or silicon (Si)[33]. This poses significant challenges due to the high two-photon absorption of III-V semiconductors, limiting the supported optical average power of femtosecond pulses to a few milliwatts for Si[34] and sub-milliwatts for InP[35] (discussion is provided in Supplementary Information Sec. 6A). Nevertheless, power levels on the order of 337 nW have been achieved with bandwidths spanning 1.5 THz[33]. Few studies offer alternative solutions to these limitations by leveraging $\chi^{(2)}$ effects on-chip in the pulsed regime, for example in hybrid Si-organic photonic circuits[36,37]. However, strong optical absorption limits the on-chip power to microwatts, restricting prior demonstrations to detection only. Consequently, broadband terahertz emission from this platform remains a challenge.

Exploiting $\chi^{(2)}$ effects in wide bandgap materials circumvents two-photon absorption and enables broadband terahertz generation and detection using femtosecond optical sources at telecom wavelengths. Furthermore, in contrast to carrier-injection techniques in semiconductors, the demonstrated large analog bandwidth of these materials could provide effective solutions to modulate terahertz radiation at high speeds, low drive voltages and with high linearity, faithfully following the optical drive signals[38].

Two key ingredients of any electromagnetic system are waveguides, used to route signals, and cavities, needed to store and amplify them. These have been successfully implemented in integrated photonics[39]. On the terahertz side, metallic transmission lines[40,41] and split ring resonators[42] can effectively route, store and filter these signals on-chip. Their sub-wavelength confinement is important to bridge the large wavelength mismatch between optics and terahertz. Apart from metallic waveguides, also dielectric waveguides[43,44] and topological waveguides have been proposed[45–47]. These building blocks have been successfully interfaced with laser-pumped photoconductive emitters and detectors to address spectroscopy applications[21,22,48,49], where the device is top-illuminated through fiber or from free-space. Developing effective methods to route and enhance terahertz waves alongside with optical signals in photonic integrated circuits presents significant challenges (Fig. 1a). These challenges arise primarily from the wavelength mismatch between optics and terahertz waves, as well as the significant material absorption and dispersion at terahertz frequencies[50]. For example, even leading wide bandgap optical platforms with high $\chi^{(2)}$ nonlinearity and robust handling of high powers reaching hundreds of milliwatts[51], such as lithium niobate, exhibit strong dispersion and significant absorption at high terahertz frequencies. For instance, at 1 THz, LN's refractive index reaches n($\omega_{THz}$) = 5 with a dispersion parameter as high as $\beta_2 = 0.5 \text{ps}^2/\text{mm}$ and an absorption coefficient of $\alpha = 10 \text{ cm}^{-1}$[52]. This is in stark contrast to optical frequencies ($\omega_p$), where the refractive index of LN is significantly lower (n($\omega_p$) = 2.1), and remains relatively flat across these frequencies[53]. In bulk systems, these contrasting material properties have complicated the generation of broadband terahertz through processes such as optical rectification. Developing a knob to

tune the terahertz refractive index by means of integrated photonics would allow to realize collinear phase matching and circumvent limitations present in bulk crystals where collinear generation is possible only in ultra-thin crystals[54,55], offering limited output terahertz power, or in periodically poled lithium niobate, limiting the bandwidth for efficient terahertz-optical conversion[56–58]. This would also provide alternatives to non-collinear techniques such as terahertz generation at the Cherenkov angle[59,60], or tilted pulse front technique[61–63] which result in a non-Gaussian terahertz beam shape. Initial demonstrations focus on patterning dielectric slab waveguides for the optical and terahertz experimentally[64] or theoretically[65]. The integration of LN rib waveguides is possible by patterning thin film lithium niobate (TFLN)[14]. Additionally patterning terahertz bowtie antennas around such rib waveguides have led to on-chip control over the emission frequency, amplitude, and time-domain waveform of the terahertz field[66]. On the downside, these configurations were limited by short interaction lengths–less than half a terahertz wavelength and long femtosecond pulses above 500 fs–which severely restricted the power and frequency range of the generated terahertz radiation (Fig. 1b, left panel)[66].

Building on these efforts, we introduce a hybrid architecture for phase-matched integrated terahertz transmission lines on TFLN, enabling broadband terahertz generation through optical rectification of sub-60 fs optical pulses (photograph in Fig. 1a) and broadband terahertz detection through on-chip electro-optic sampling. Our device consists of a rib waveguide inside a terahertz transmission line to generate, couple, guide and detect terahertz fields effectively (Fig. 1b, center panel). This confines the terahertz to a strongly sub-wavelength cross section ($\frac{S_{eff}}{\lambda_{THz}^2} = 10^{-5}$), thereby mitigating the associated dispersion and absorption losses. By providing short pump pulses to our devices, the spatial extension of the pulse becomes shorter than the transmission line and phase matching with the terahertz pulse is required for high efficiency. We demonstrate the design's ability to achieve phase matching across an extensive frequency range from 100 GHz up to 3.5 THz (Fig. 1c) with a coherence length with values of few millimeters (Fig. 1d) and obtain spatial overlap of the terahertz and optical modes (insets Fig. 1c). Altogether, this enables the generation of a broadband terahertz pulse with frequencies spanning from 200 GHz up to 3.5 THz through optical rectification, and a field amplitude higher than previous work in TFLN by two orders of magnitude (Fig. 1b, middle panel)[66]. Additionally, the incorporation of terahertz transmission lines into our photonic circuit offers a high degree of control over the terahertz electric field. The open-ended transmission line has a reflection coefficient exceeding 90%, allowing us to realize a strip line terahertz cavity. Routing optical signals into the cavity provides means to generate terahertz modes at discrete frequencies that can be outcoupled into the farfield by an antenna (Fig. 1b, right panel). Likewise, our photonics-integrated transmission lines support detection of broadband terahertz radiation up to 3 THz through electro-optic sampling, featuring a 60 dB dynamic range (in power) at 100 ms integration time per point, surpassing prior demonstrations[36,37] in bandwidth, signal-to-noise ratio and modulation strength. Implemented on the same LN substrate, this lays the foundation for integration of both components on the same chip. Examples of practical applications that may adopt these concepts include enhancing the speed and resolution of millimeter-wave radar systems[67], implementation of Kerr combs for terahertz communications[68], or terahertz imaging using photonics[69]. This integration not only aligns with modern fabrication standards but also opens new avenues for chip-scale applications in telecommunications, spectroscopy, and computing.

## Results
### Broadband photonics-integrated terahertz emitters
Terahertz generation occurs when a short pulse of light undergoes optical rectification (OR) as it travels through a medium with a non-vanishing second-order nonlinearity $\chi^{(2)}$. This nonlinear process leads to the generation of a second-order polarization P$^{(2)}$(t) following the

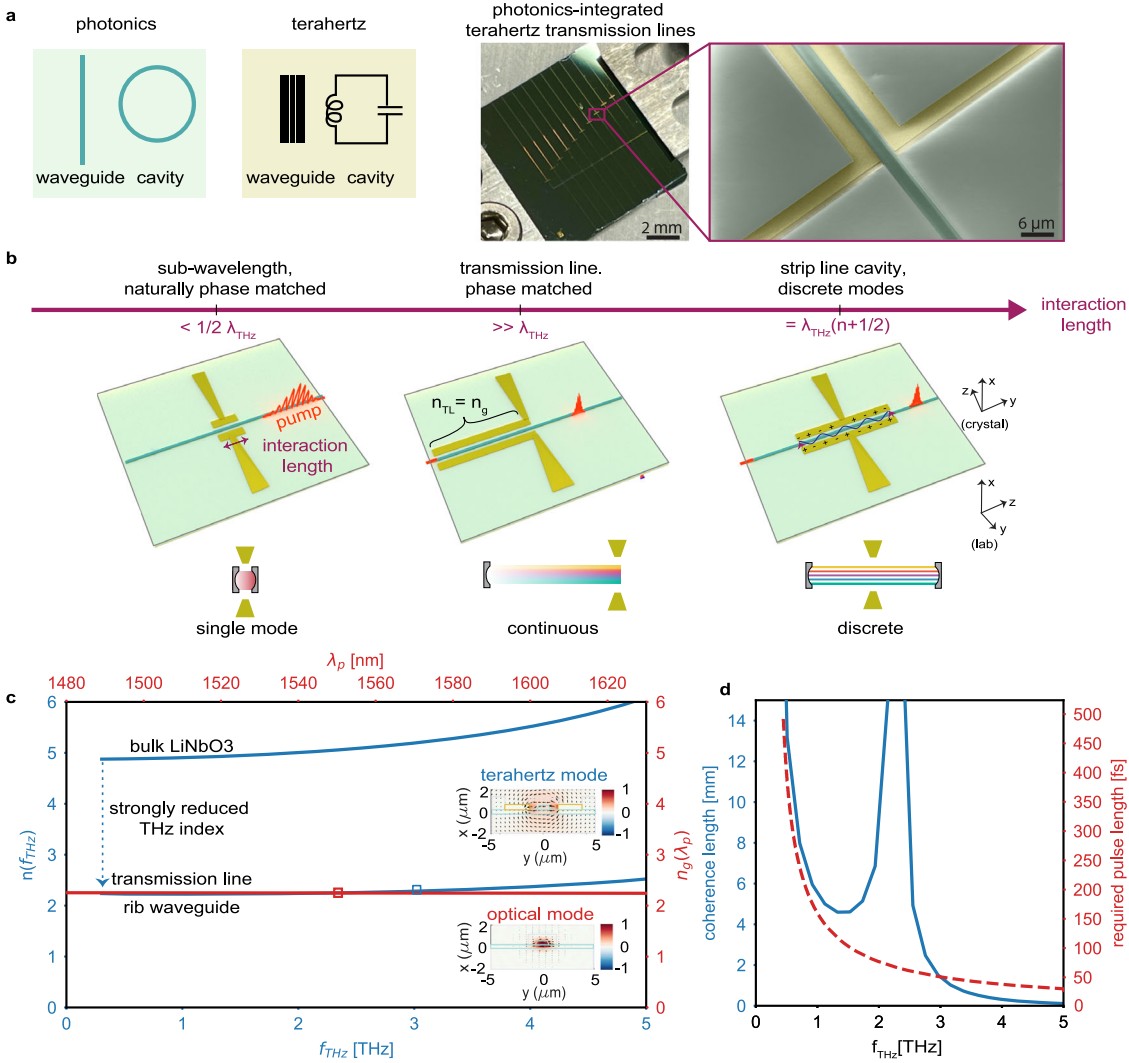

**Fig. 1 | Integration of terahertz and photonics on a single chip. a** Waveguides and cavities on chip for the optical and the terahertz regime. Photograph and false color scanning electron microscope (SEM) picture of our proposed chip that integrates photonics with terahertz transmission lines. Picture shows cladded LN waveguides (dark green in the SEM) and antenna-coupled terahertz transmission lines (yellow). **b** Efficient bidirectional conversion between the optical and the terahertz waves for generation and detection requires matching the terahertz effective index $n_{TL}$ with the group index $n_g$ of a telecom pulse propagating inside a rib waveguide. The pulse length corresponds to a spatial envelope inside transmission line and phase matching becomes essential when this envelope becomes shorter than the transmission line length. In addition, an antenna can be patterned at the end of the transmission line to facilitate in-/out-coupling of the terahertz.

Alternatively, the terahertz transmission line can be terminated with open ends on both sides to realize strip line cavities that support a discrete set of modes (on-resonance $l_{TL} = (n + \frac{1}{2})\lambda_{THz}$, right panel). **c** The effective refractive index of the fundamental mode at terahertz frequencies of the transmission line is matched to the group refractive index of the rib waveguide at telecom frequencies, while preserving good spatial overlap (inset). **d** Efficient generation/detection of a specific terahertz frequency $f_{THz}$ is achieved by satisfying two conditions. First, our phase matching enables coherence lengths larger than a few millimeters in a collinear, co-propagating configuration (left axis). Second, our optical pump has a duration that is sufficiently short to efficiently generate/detect the corresponding frequency (right axis).

intensity envelope of the optical pulse I(t), which acts as a source of the emitted terahertz radiation. Generally, the bandwidth of the generated terahertz field is limited by the temporal duration of the optical pulse, while its amplitude scales linearly with the input pulse's intensity. In our case, the terahertz wave is generated inside the TFLN waveguide and it propagates along a transmission line (realized with gold strip lines) defined along the waveguide (see Fig. 1b). The exact geometry is detailed in the Supplementary Information Sec. 1A. The generated terahertz pulse depends on three factors. First, femtosecond short optical pulses are needed to maximize the intensity envelope, thereby enhancing the generation efficiency and the bandwidth. This is typically complicated by the presence of dispersive components in the pulse delivery setup to the terahertz emitter. Second, it is essential to fulfill phase matching between the

terahertz and optical waves over the entire bandwidth offered by the femtosecond pulse. This would achieve maximal terahertz bandwidth. Finally, we need to outcouple the broadband terahertz wave into the farfield.

To accomplish an efficient delivery of ultra-short pulses generated by a mode-locked laser, centered at wavelength of 1550 nm (repetition rate of 100 MHz), we use an edge-coupling (butt-coupling) approach (Fig. 2a). Furthermore, the dispersion is controlled using a prism pair and dispersion compensating fiber (DCF). To further mitigate the losses of the various components and achieve short pulses at the chip by self-phase modulation inside the fiber, an Erbium-doped fiber amplifier (EDFA) is employed before coupling to the chip. Our dispersion compensated setup generates pulses as short as 60 fs, corresponding to a 10 dB optical bandwidth of 75 nm or 9.2 THz (see Fig. 5a

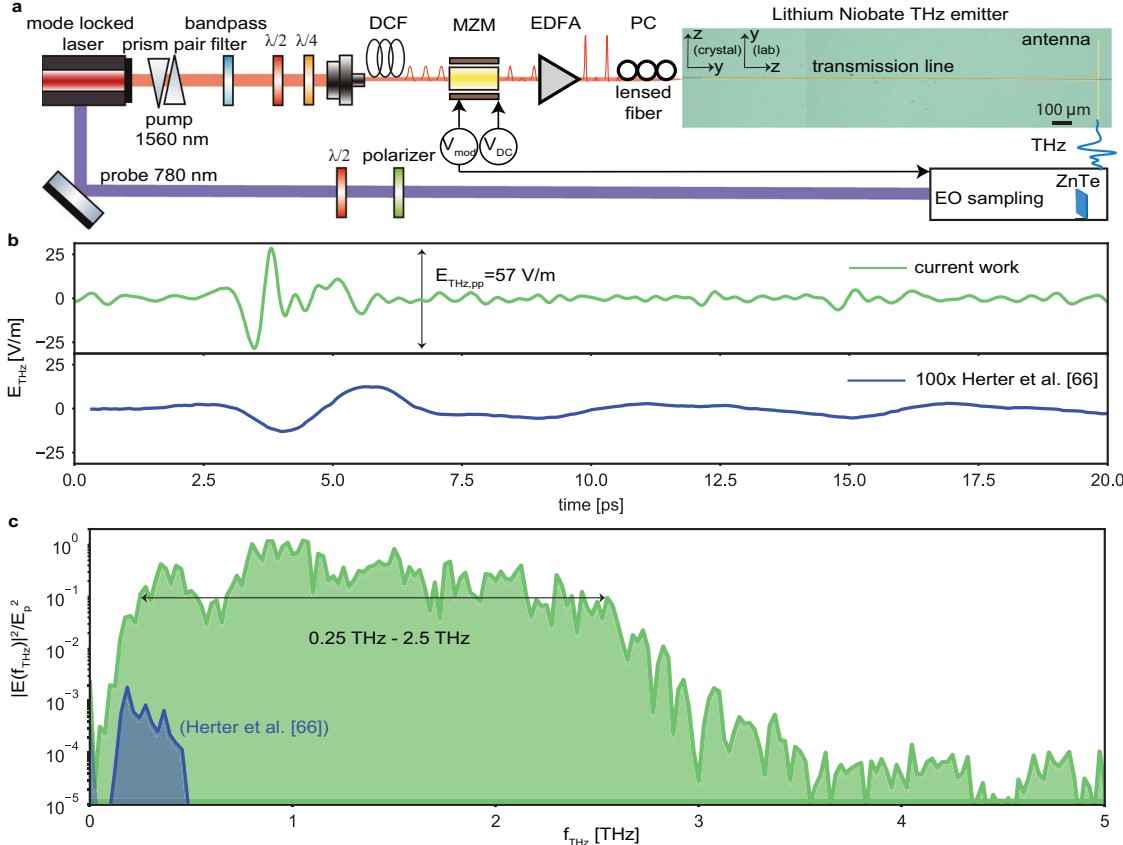

**Fig. 2 | Photonics-integrated broadband terahertz emitter. a** Optical setup used for the characterization of the terahertz emitters. Modulated optical pump pulses are coupled via tapered fibers into the rib waveguide. Dispersion compensation in fiber (DCF) ensures their compression at the input of the chip. A commercial electro-optic (EO) Mach-Zehnder modulator (MZM) controls the intensity of these pulses by means of applied bias voltages $V_{DC}$ and $V_{mod}$. Broadband terahertz radiation is generated inside the transmission line, and gets emitted into the farfield by terminating the structure with a broadband multi-wave antenna. This is detected using free-space electro-optic sampling inside a zinc telluride crystal. **b** Electric field

of the TFLN emitter, featuring a transmission line of length $l_{TL} = 2$ mm, terminated by a dipolar antenna of length $l_{ant}$ on one end, as-measured inside a 1 mm zinc telluride (ZnTe) crystal in comparison with our last demonstration (100 × , lower panel). **c** The corresponding spectra is retrieved by taking the Fourier transform of the time traces, showing a dynamic range of 50 dB in intensity, and a maximum emission frequency of $f_{max} = 3.5$ THz, clearly outperforming our previous work[66]. Both spectra are normalized to the pump pulse energy on-chip, which in our case is 100 pJ. *EDFA* erbium doped fiber amplifier, *PC* polarization controller.

and b in Supplementary Information Sec. 1C). In contrast to previously used grating couplers[66,70,71], edge couplers allow for coupling the entire bandwidth of the optical pump into the TFLN waveguides (see comparison of edge coupling and grating couplers in Supplementary Fig. 5c in). To benefit from the largest component of the nonlinear tensor $\chi^{(2)}_{333}$, we align the polarization of the optical pulse to the z-direction of the TFLN.

To achieve phase matching between the terahertz and optical waves, we match the group velocity of the optical wave with the phase velocity of the terahertz wave by engineering the geometry of the transmission line, similar to the design concept used in broadband electro-optic modulators. The partial extension of the guided teraahertz mode outside the LN material lowers its effective refractive index to $n_{TL} = 2.25 - 2.3$ thereby matching it with the that of the pump (experimentally determined to be $n_g = 2.25$, see Supplementary Information Sec. 1B). Phase-matching leads to a linear increase of the terahertz field over the entire length of the transmission line $l_{TL}$. In the frequency domain, the terahertz electric field at the end of the transmission line is given by:

$$\tilde{E}_{THz}(l_{TL}, \omega_{THz}) = \frac{i\pi\chi^{(2)}E_0^2\omega_{THz}^2\tau^2 l_{TL}\Gamma_{OR}}{4c_0 n_{TL}(\omega_{THz})\sinh(\pi\omega_{THz}\tau/2)}G_{TL}(\omega_{THz}) \cdot e^{-i\frac{\omega_{THz}l_{TL}}{c_0}n_{TL}(\omega_{THz})} \quad (1)$$

where $E_0$ is the pump electric field amplitude, $\tau$ is the FWHM of the pump pulse, $c_0$ is the speed of light in vacuum, $\chi^{(2)} = \chi^{(2)}_{333} = 360$ pm/V is the second-order susceptibility along which all fields are oriented[72], and $n_g$ is the pump group index (Supplementary Information Sec. 2A). The optical losses are negligible over the transmission line length we consider here. We introduce the phase-matching function that accounts for phase mismatch and terahertz losses:

$$G_{TL}(\omega_{THz}) = \frac{e^{-i\Delta k l_{TL}} - e^{-\alpha l_{TL}/2}}{i\Delta k l_{TL} - \alpha l_{TL}/2} \quad (2)$$

with $\Delta k = \frac{\omega_{THz}}{c_0}(n_g - n_{TL}(\omega_{THz}))$ being the wave vectors mismatch and $\alpha$ the loss coefficient of the guided terahertz field, defined as $E_{THz}(z) = E_{THz}(0)e^{-\alpha z/2}$. The coherence length of our transmission line is 5 mm at 1 THz and 0.3 mm at 4 THz (Fig. 1d). The phase matching function can be recast into an effective interaction length

$$l_{eff}(\omega_{THz}) = l_{TL}G_{TL}(\omega_{THz}) \quad (3)$$

that depends on the terahertz frequency and quantifies the length over which the terahertz is effectively generated, given the losses and the phase mismatch present in the system. The amount of generated teraahertz electric field that couples to the transmission line is also

dependent on the overlap factor:

$$\Gamma_{OR} = \frac{\iint_{(x,y)} g_{opt}^2(x,y) g_{THz}(x,y) dx dy}{\iint_{(x,y)} |g_{THz}(x,y)|^2 dx dy} \qquad (4)$$

where $g_{THz}(x,y)$ and $g_{opt}(x,y)$ are the normalized spatial distributions of the THz and optical modes as visualized in the inset of 1c, respectively. Using finite element method simulations (CST Microwave Studio), we find that the overlap $\Gamma_{OR} = 0.1$ depends weakly on frequency in the entire bandwidth spanning from 100 GHz to 4 THz (see Supplementary Information Fig. 9c).

We note that similar to the RF loss in modulators, the terahertz loss of the transmission line and the terahertz-optical mode overlap limit the bandwidth of the generated spectrum. In particular, both the radiative loss $\alpha_{rad}$ (leaking of the terahertz wave through the substrate) and the absorption loss $\alpha_{abs}$ (where terahertz field is absorbed by both gold and TFLN) impact the generation efficiency at higher frequencies more severely (Supplementary Information Fig. 8). These loss values and the phase matching condition can all be optimized by tuning the dimensions of the transmission line (Supplementary Information Sec. 2C). Following the full analysis, we find that $l_{TL} = 2$ mm provides phase-matching for broadband emission up to a frequency of $\omega_{THz} = 2\pi \times 3.5$ THz. We terminate the transmission line with a dipolar antenna with $l_{ant} = 200\,\mu m$ and width $w_{ant} = 5\,\mu m$, designed for broadband response. We find that these antenna dimensions support emitting a broad terahertz spectrum perpendicularly to the chip, by operating on the various higher-order modes (Supplementary Information Sec. 2G). We note that the beam shape of the emitted terahertz strongly depends on the terahertz frequency and antenna dimensions and can deviate from a Gaussian shape.

We test our TFLN emitter experimentally, by using the electro-optic sampling technique to recover its temporal waveform (a description of the entire experimental setup is given in Supplementary Information Sec. 1C). The measured terahertz signal is shown in Fig. 2b (upper panel, green line), featuring an electric field peak-to-peak of $E_{THz,pp} = 57$ V/m (as measured inside the zinc telluride detection crystal) showing a ~100 fold

improvement in field amplitude over our previous work (lower panel, blue line)[66] and a significantly reduced duration of the terahertz pulse. The measurement represents a single time trace, acquired with a step size of 25 fs and an integration time of 100 ms per time point. The frequency spectrum is obtained by taking the Fourier transform of the temporal waveform revealing a broad and relatively flat-top emission, approaching a maximal frequency of 3.5 THz (green spectrum in Fig. 2c, exhibiting a 10 dB bandwidth of ~2.5 THz). The maximal frequency of 3.5 THz is in good agreement with the calculated coherence length. Compared to our previous work (blue spectrum in Fig. 2c)[66], we recorded a four-octave spanning spectrum with a dynamic range of ~50 dB in intensity, which makes our platform suitable for applications in spectroscopy requiring a broad bandwidth. We experimentally compare various transmission line lengths of $l_{TL} = 0.12$, 0.5 and 2 mm and find that a long transmission line benefits the generation efficiency of low terahertz frequencies compared to shorter transmission line (Supplementary Information Sec. 3C).

We note that outside the transmission line, the strong terahertz-optical mismatch supports efficient generation only at a an angle of $\theta_c = 42°$ with the yz-plane into the high-refractive index Si substrate (see CST simulations in Supplementary Information Sec. 2F). By accounting for reflections (Fresnel and total internal reflection at angles beyond the critical angle) at the high-resistivity Si substrate, at the detection crystal, limited numerical aperture of our detection system and asymmetric emission into the dielectric below and above the chip, we estimate the terahertz electric field at the end of the transmission line to be approximately a factor of $10^4 - 10^5$ higher than the measured one and amount to $E_{THz,TL} = 10^5 - 10^6$ V/m (see Supplementary Information Sec. 2G and 3A). This agrees with our full-field model which accounts for all losses, phase matching, and spatial overlap of the mixing fields (see estimated fields in Supplementary Information Fig. 11).

## Electro-optic control of the terahertz field amplitude

Amplitude modulation is one of the cornerstones of current communication schemes. Precise control over the amplitude of the generated terahertz fields is essential for future generations of terahertz devices.

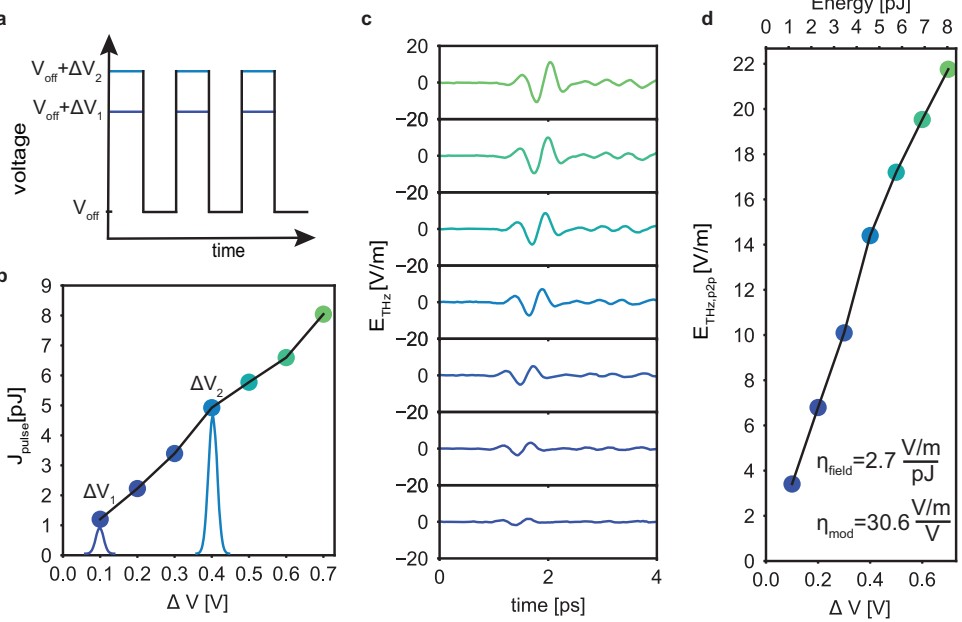

**Fig. 3 | Electro-optic control of the terahertz field amplitude. a** The pump intensity modulation scheme used for THz amplitude control. The pump is switched between an off-state at $V_{off}$, where no pump pulses reach the lithium niobate terahertz emitter, and an on-state $V_{off} + \Delta V$ with controllable pump pulse energy. In the case of our commercial Mach-Zehnder modulator the off-state voltage is $V_{off} = 6$ V. **b** Dependence of the pump pulse energies inside the TFLN chip on the applied modulation voltage $\Delta V$ shows that energies up to $J_{pulse} = 8$ pJ depend linearly on applied voltages up to $\Delta V = 0.7$ V. **c** measured terahertz electric fields for the various control voltages $\Delta V$. **d** Peak-to-peak values of the terahertz electric field $E_{THz,pp}$ exhibit linear dependence on $\Delta V$. We measure a field generation efficiency of $\eta_{field} = \frac{E_{THz,pp}}{J_{pulse}} = 2.7\,\frac{V/m}{pJ}$ and a field modulation efficiency of $\eta_{mod} = \frac{E_{THz,pp}}{\Delta V} = 30.6\,\frac{V/m}{V}$.

Equation (1) reveals the numerous knobs that could be used, such as changing the nonlinear crystal thickness (interaction length), changing the input pulse duration, or the overlap between the terahertz and optical field. Among these possibilities, controlling the amplitude of the input femtosecond pulse is the most widely used technique, typically implemented using a mechanical chopper. Chopping the input beam also aids the lock-in detection scheme towards recording the terahertz waveform. However, the slow speed of mechanical choppers (kHz rates) compromises the signal to noise ratio (SNR) since flicker noise scales with $\propto 1/f$. More importantly, this technique provides only an on-off modulation without the ability to continuously control the amplitude of the input

pulse. Even such an on-off modulation exhibits unfavorable performance due to slow rise and fall times, consequently leading to a worsened signal-to-noise ratio. To achieve full control over the generated terahertz amplitude without these issues, we implement a fiber-pigtailed Mach-Zehnder electro-optic (EO) intensity modulator before our chip.

We adopt this EO control on a shorter device compared to the one in Fig. 2 ($l_{TL} = 120\,\mu m$ vs $l_{TL} = 2\,mm$) which features a temporal profile closer to a single cycle. We operate the modulator by applying a square wave voltage at 1 MHz (below the repetition rate of the laser), toggling between an off-state and an on-state where a desired portion of the pump pulse energy is sent into the chip (Fig. 3a). By adjusting the

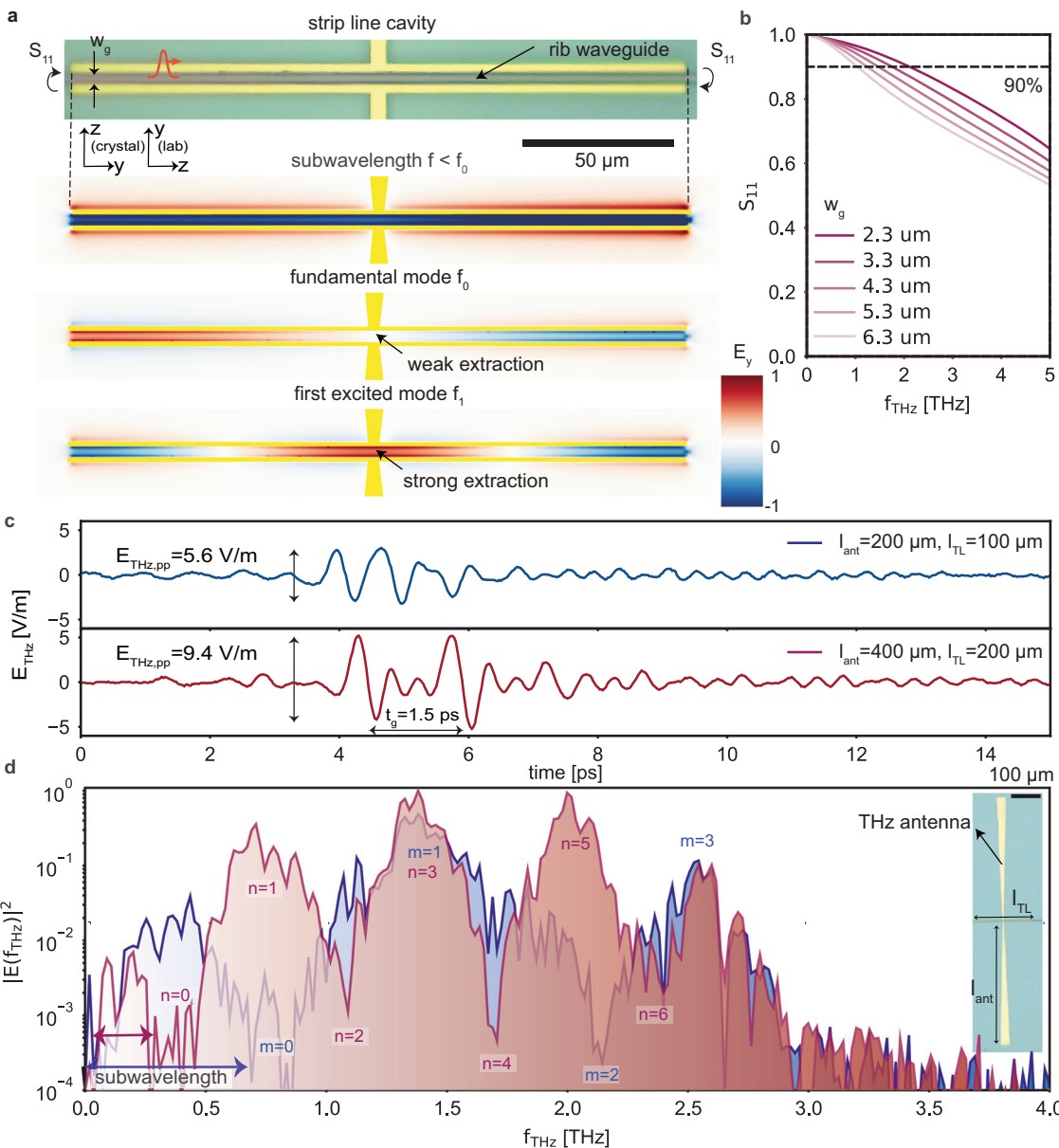

**Fig. 4 | Strip line terahertz cavities from transmission lines. a** A terahertz cavity is formed by terminating a transmission line with an open end on both sides. We excite the cavity modes by optical rectification of pump pulses that travel along the rib waveguide. To access these cavity modes, we pattern a broadband antenna centrally around the transmission line. **b** The reflection coefficients of the open termination reaching above $S_{11} > 90\%$ at frequencies below 1.7 THz and $S_{11} = 78\%$ at 3 THz, from CST simulation. These values depend on the sub-wavelength dimension of the cross-section of the transmission line. **c** Experimental results of the terahertz field emitted from different strip line cavities with length of $l_{TL} = 100\,\mu m$ and

$l_{TL} = 200\,\mu m$. Clearly visible, especially in the case of the longer cavity, is a train of pulses which in that case exhibits a group delay of $t_g = 1.5$ ps. The THz field amplitude of the initial terahertz pulse is approximately twice larger for the case of the twice longer strip line cavity ($E_{THz,pp,1} = 9.4$ V/m vs $E_{THz,pp,1} = 5.6$ V/m), in line with expectation due to the ratio of the interaction lengths. **d** The Fourier transforms of the terahertz electric field reveal broadband spectra that feature discrete modes, experimentally confirming the build-up of cavity modes. Indeed, the twice longer strip line cavity features twice as many modes compared to the shorter one with resonances that align well in frequency.

amplitude of the ON state ($\Delta V$), we send various input pulse energies $J_{pulse}$ (Fig. 3b) resulting in a range of electric-field amplitudes (Fig. 3c). Consequently, we find that the corresponding peak-to-peak electric field $E_{THz,pp}$ of the emitted terahertz electric field exhibits a linear dependence on the modulated pump pulse energy and, consequently, on the applied voltage $\Delta V$ (Fig. 3d). Analysing the slope, we find a generation efficiency of $\eta_{field} = \frac{E_{THz,pp}}{J_{pulse}} = 2.7\,\frac{V/m}{pJ}$ and a field modulation efficiency of $\eta_{mod} = \frac{E_{THz,pp}}{\Delta V} = 30.6\,\frac{V/m}{V}$. We further analyze the noise properties of our setup. We find that for a modulation frequency of 1 MHz, the noise of our lock-in detection is slightly above the shot-noise limit (the measured voltage fluctuation is $\delta V_{meas} = 160$ nV while the

expected shot-noise limited voltage fluctuation is $\delta V_{shot\,noise} = 148$ nV, see Supplementary Information Sec. 3B). This experimentally confirms the advantages of MHz-speed electro-optic modulation of the input pulse for reaching low noise at the fundamental shot noise limit.

## Terahertz cavities from coplanar transmission lines

Apart from the transmission lines, integrated cavities are crucial to expanding the capabilities of terahertz integrated photonic devices. They serve important functionalities such as temporarily storing light, spectral filtering, and field enhancement dictated by the their facet reflectivity[73]. One way to realize a strip-line cavity would be to leave both sides of a transmission line open-ended (unterminated), which

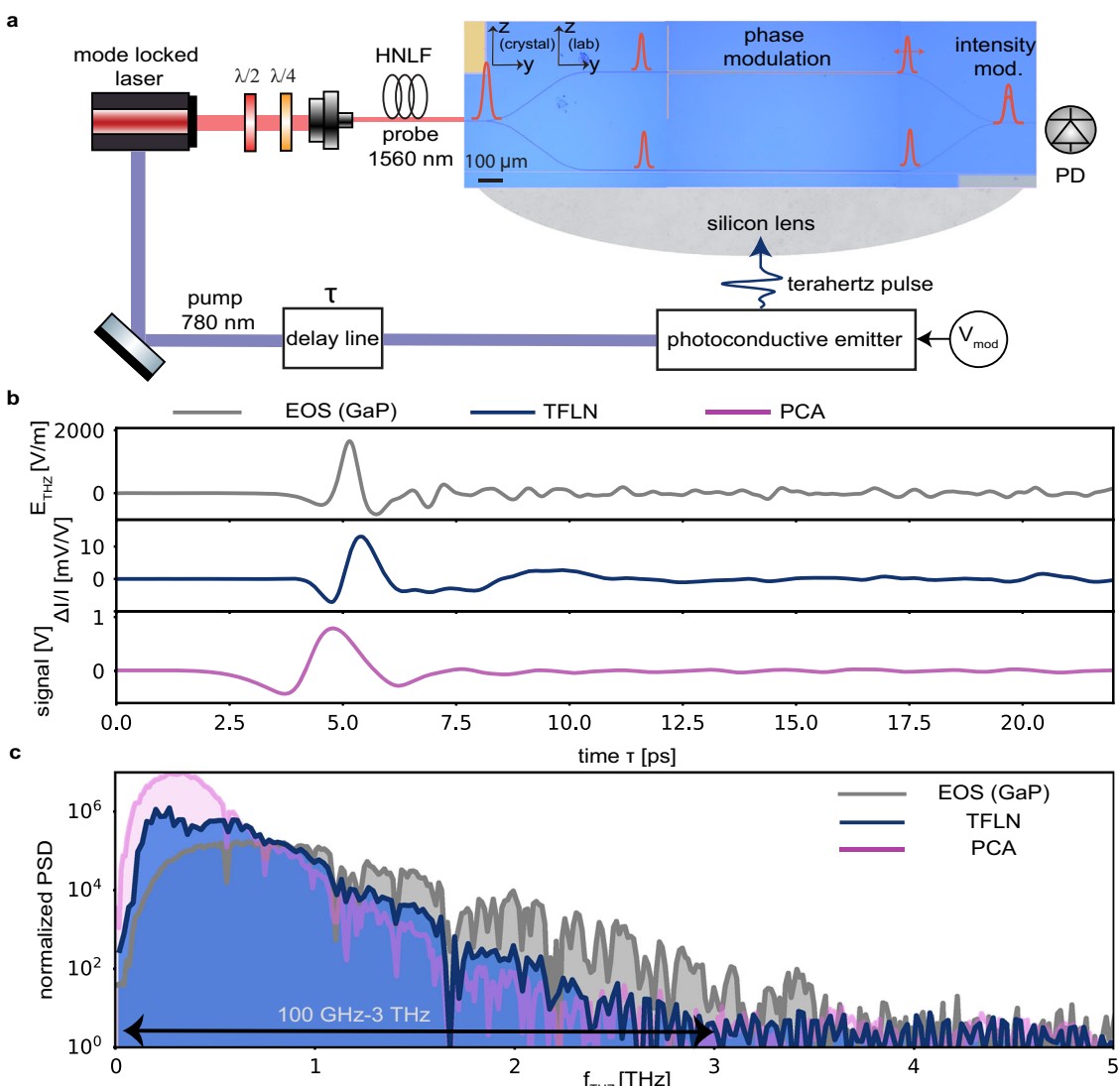

**Fig. 5 | Photonics-integrated broadband terahertz detector. a** Optical setup used for the characterization of the terahertz detector. The detected terahertz radiation is emitted from a commercial GaAs-based photoconductive antenna pumped at 780 nm. The probe at 1560 nm is coupled to a highly nonlinear fiber (HNLF) to compress the pulse length down to 60 fs. Broadband terahertz detection in TFLN is implemented through on-chip electro-optic sampling. In electro-optic sampling, a terahertz electric field modulates the phase of a femtosecond probe. Measuring this phase allows reconstructing the incident terahertz signal. In our implementation, the probe pulse is split in power to propagate along two arms of a Mach-Zehnder interferometer. In one arm, a dipole antenna captures the terahertz radiation which propagates inside the transmission line to provide phase modulation to the probe pulse via the electro-optic (Pockels) effect. The phase-modulated probe propagates further to the output splitter of the interferometer,

where it interferes with the probe that propagated along the second arm. This interference results in an intensity modulation at the output of the interferometer that is measured with a photodiode (PD). **b** The upper panel shows the terahertz waveform retrieved through electro-optic sampling (EOS) in a bulk gallium phosphide (GaP) crystal which is 200 $\mu$m thick and has a flat frequency response. This allows accurate reconstruction of the input terahertz pulse. The middle panel shows the normalized modulation of the probe power from our thin-film lithium niobate (TFLN) device with the length $L_{ant} = 200\,\mu$m and a transmission line length $l_{TL} = 1$ mm. The lower panel shows the terahertz signal as-detected with a commercial photoconductive antenna (PCA). **c** Fourier transforms of the signals in (**b**) compares the various detection schemes, showcasing the ability of our TFLN chip to reach 3 THz bandwidth with 60 dB dynamic range.

would constitute a strip line cavity[74]. A given cavity mode would either have a node (for odd modes) or an anti-node (for even modes) at the center of the cavity strip line. Therefore, an antenna placed at the center can extract the odd modes into the far field (strong extraction), while not efficiently out-coupling even modes into the far-field (weak extraction) (Fig. 4a). Since the transverse dimensions of our transmission lines are highly sub-wavelength compared the generated terahertz wavelength, the reflection coefficient exceeds $S_{11} = 79\%$ in the entire band up to $f_{THz} = 3$ THz (see the $S_{11}$ in Fig. 4b). We fabricate two strip-line cavities with lengths of $l_{TL} = 100\,\mu m$ and $l_{TL} = 200\,\mu m$ to have different mode profile at the antenna. We selected these two candidate geometries to evaluate the temporal and spectral behavior of the generated field (circulating inside the cavity) with respect to strip-line length and antenna dimensions (Fig. 4c).

As the optical pump propagates inside the transmission line and the terahertz radiation accumulates, a portion of the terahertz field is coupled to the antenna, leading to the initial terahertz pulse formation. The remaining terahertz field continues to build up until the optical pump reaches the end of the transmission line and the terahertz wave reflects back. The optical pump leaves the structure, and the reflected terahertz radiation experiences no more parametric gain during propagation. Once the terahertz pulse reaches the antenna again, it is coupled out with a group delay of $t_g = \frac{n_g l_{TL}}{c}$ with respect to the initial terahertz pulse. This cycle repeats until the terahertz field has been entirely coupled out into the far field, absorbed in the lossy materials, or leaked out through the cavity mirrors. The build-up in the shorter transmission line is slightly weaker in terahertz field ($E_{THz,pp,1} = 5.6$ V/m) compared to the longer one ($E_{THz,pp,2} = 9.8$ V/m) due to shorter interaction length. However, its group delay $t_g = 0.7$ ps is comparable to the duration of the terahertz pulse, in contrast to the case of the longer cavity. Consequently, the terahertz field is stored inside the cavity for shorter times (~4 ps) as opposed to the longer cavity, where it exceeds 6 ps, showcasing how cavities can extend the light lifetime inside a cavity.

The Fourier Transforms of these time-domain traces reveals formation of discrete modes with their exact frequency depending on the length of the transmission line (Fig. 4d), unlike the broadband spectrum of the traveling wave emitter in Fig. 2c. The spectral analysis of the terahertz emission reveals two key findings. Firstly, the fundamental cavity mode ($n = 0$) is not extracted by the antenna due to its placement at the minima of the cavity modes. Secondly, at frequencies below the fundamental mode, the antenna operates in the Hertzian dipole regime with sub-wavelength interaction length (which was explored in Ref. 66). This regime is characterized by a uniform distribution of the terahertz field showing no visible z-dependency (Fig. 4a, sub-wavelength $l_{int} < \frac{\lambda_{THz}}{2}$). Our full cavity model shows that the quality factor is limited by the losses inside the terahertz transmission line. The model captures the experimental details such as field amplitudes in the time domain, spacing of the measured resonances in the frequency domain, and their linewidth (full details in Supplementary Information Sec. 2D).

## Broadband photonics-integrated terahertz detectors

To establish a versatile and scalable terahertz platform, both generation and detection must be realized on the same integrated photonic chip. Here, we show that our proposed antenna-coupled transmission lines can be reciprocally used to achieve efficient detection of broadband terahertz pulses via the on-chip electro-optic sampling (Fig. 5a). The on-chip antenna collects the terahertz signal that is incident from the rear of the chip and directs it into the transmission line. Similar to RF phase modulators, the femtosecond probe pulse with a center frequency $\omega_{pr}$ propagating along the optical waveguide undergoes a phase modulation when interacting with the terahertz pulse[70]. The degree of phase modulation experienced by the probe pulse depends on the instantaneous terahertz electric field $E_{THz}(x, y, z, t)$. Delaying the terahertz pulse by a time $\tau$ with respect to the probe pulse by means of

a delay stage allows to reconstruct the incident terahertz electric field by measuring:

$$\Delta\phi(\tau) = \frac{\omega_{pr}}{c_0 n(\omega_{pr})} \chi^{(2)} \Gamma_{EO} l_{TL} E_{THz} \frac{1}{2\pi} \int_{-\infty}^{\infty} \tilde{\varepsilon}_{THz}(\omega_{THz}) G_{TL}(\omega_{THz}) e^{-i\omega_{THz}\tau} d\omega_{THz} \quad (5)$$

where $\Gamma_{EO}$ is the overlap factor defined as follows:

$$\Gamma_{EO} = \frac{\iint_{(x,y)} g_{opt}^2(x,y) g_{THz}(x,y) dx dy}{\iint_{(x,y)} |g_{opt}(x,y)|^2 dx dy}, \quad (6)$$

$E_{THz}$ is the incident terahertz field amplitude and $\tilde{\varepsilon}_{THz}(\omega_{THz})$ describes the double-sided complex spectrum of the terahertz pulse at the beginning of the transmission line $z = 0$. Using finite element method simulations (CST Microwave Studio), we find that the overlap $\Gamma_{EO} = 0.6$ depends weakly on frequency in the entire bandwidth spanning from 100 GHz to 4 THz (see Supplementary Information Fig. 21). The efficiency of the interaction depends on the phase matching function $G_{TL}(\omega_{THz})$, which is the same as for terahertz generation (see full derivation in Supplementary Sec. 4A). Eq. (5) shows that the bandwidth of the detected terahertz field is limited by the phase mismatch between the optical and the terahertz field, terahertz losses, the strength of the nonlinear interaction $\chi^{(2)}$, and the overlap integral. As with generation, terahertz detection benefits from incorporating antenna-coupled terahertz transmission lines with the optical waveguide because it achieves phase-matched co-propagation of terahertz and optical signals over extended lengths and bandwidths up to 3.5 THz. This allows to significantly increase the interaction length $l_{TL}$ and thereby the accumulated phase $\Delta\phi(\tau)$, overcoming limitations of previous reports[36,70,75], which lacked phase-matching and necessitated sub-wavelength interaction lengths. To detect the entire bandwidth of the incoming terahertz pulse, it is essential to accomplish an efficient delivery of ultra-short probe pulses. Similar to the case of terahertz emission, we couple 60 fs telecom pulses through the edge of the chip, ensuring ultrashort pulses inside the transmission line. Details about the measurement setup are given in Supplementary Information Sec. 1D. To facilitate the read-out of the terahertz electric field, we embed the transmission line into one arm of a Mach-Zehnder interferometer, similar to previous works[36,70,75]. The probe pulses are thus split using a y-combiner and propagate further along the two arms of the interferometer. After experiencing phase modulation in one of the two arms of the interferometer, the two beams are recombined at its output. Similar to RF intensity modulators, this transforms the phase modulation into an intensity modulation that is linearly proportional to the terahertz electric field since $\frac{\Delta I_{out}(t)}{I_{out}} = \Delta\phi(t)$. Taking the Fourier transform of this intensity modulation retrieves the power spectral density (PSD) of the terahertz signal detected by our TFLN chip:

$$PSD(\omega_{THz}) = |\mathcal{F}\{\frac{\Delta I_{out}(\tau)}{I_{out}}\}|^2 = \frac{\omega_{pr}^2}{c_0^2 n(\omega_{pr})^2} \chi^{(2)2} E_{THz}^2 \Gamma_{EO}^2 l_{eff}(\omega_{THz})^2 \tilde{\varepsilon}_{THz}(\omega_{THz})^2 \quad (7)$$

We test our TFLN detector by shining onto our device a terahertz pulse emitted from a commercial photoconductive emitter from low temperature GaAs, pumped by a femtosecond pulse centered at a wavelength of 780 nm. We calibrate the strength of the input terahertz signal by performing electro-optic sampling with a 200 $\mu m$-thick GaP crystal, which provides phase matching beyond 5 THz when probed at 1550 nm[76], and find an input terahertz amplitude of 16.5 V/cm, shown in gray in Fig. 5b, upper panel. With our TFLN chip, we measure a single-cycle terahertz pulse and peak relative modulation of $\frac{\Delta I_{out}(t)}{I_{out}} = 1.3\%$ (blue curve in Fig. 5b, middle panel). We achieve this modulation efficiency by placing a Si lens on the back side of the chip. We compare the performance with and without the chip in the Supplementary

Information Sec. 5A. We use 25 fs stepsize, 100 ms integration time per measurement point, and a probe power of 13.7 $\mu$W at the photodiode. We compare the response of our chip to a commercially available InGaAs photoconductive detector (PCA) with similar antenna dimensions (pink curve in Fig. 5b, lower panel) and find that the response of our TFLN chip is faster, exhibiting a larger bandwidth compared to the PCA. This is confirmed when taking the Fourier transform of the corresponding temporal waveforms, obtaining 61 dB dynamic range at 100 ms integration time and a maximum resolvable frequency up to 3 THz for the TFLN chip (blue spectrum in Fig. 5c), which exceeds the PCA, limited to about 2 THz with 67 dB dynamic range at 300 ms integration time (Fig. 5c, pink spectrum). Although we use ~76 times lower optical probe powers and three times shorter integration time for the TFLN measurement compared to free-space electro-optic sampling in GaP (Fig. 5c, gray spectrum), we find that TFLN achieves higher dynamic range. At low frequencies, the TFLN performs better whereas the detection with GaP has a larger bandwidth, reaching up to 4 THz. In the case of TFLN, we attribute the drop in detected power with increasing terahertz frequency to two concurrent effects. First, the emission of our terahertz source drops strongly at frequencies beyond 1 THz, as confirmed by the spectrum of the input pulse (gray). And second, the terahertz losses and phase mismatch above 3.5 THz limit the effective interaction length at increased terahertz frequencies. Altogether, this explains the faster roll-off of TFLN (25.1 dB/THz) compared to the input pulse (14.4 dB/THz). This observation is further consolidated when comparing the achieved bandwidth for transmission lines of various lengths (see Supplementary Information Sec. 5B), showing that low frequencies benefit the most from long transmission lines. A short transmission line length $l_{TL}$ = 125 $\mu$m supports a bandwidth larger than 3 THz and features a roll-off of around 13 dB/THz, aligning with the detection with GaP. The noise equivalent field for the TFLN measurement is $E_{NEF}$ = 0.683 V/m. This field is commensurate with ~0.1 photons per terahertz pulse, setting the lower limit for our detectors. All measurement configurations are given in the Supplementary Information Sec. 6C. The performance of our chips is currently limited by the total in- and out-coupling of the optical probe to the chip which is below 0.1%. To improve this efficiency, it is essential to achieve probe powers at the photodetector which are on-par with free-space electro-optic sampling, and could improve the dynamic range by 20 dB.

Our results outperform prior on-chip broadband terahertz detectors in all important figures of merit, such as dynamic range, terahertz bandwidth and modulation efficiency[36,70,75], verifying the essential role of phase-matched transmission lines for efficient terahertz detection. Our detectors are fabricated on x-cut TFLN, as our emitters, allowing fabrication of both in the same run, minimizing overhead, and paving the way for more advanced terahertz systems.

## Discussion

In summary, we demonstrate that the hybrid integration of terahertz transmission lines with TFLN waveguides and dipole antennas enables compact and efficient solutions for the generation and detection of broadband terahertz radiation. First, we demonstrate broadband and spectrally flat terahertz generation up to 3.5 THz, spanning four octaves. The pump pulses are provided by an off-chip commercial mode-locked laser connected to a telecom fiber system and the detection is done with conventional electro-optic sampling. We achieve this bandwidth by overcoming the significant limitation of phase mismatch between the optical and terahertz signals, which previously caused their early walk-off in bulk. Similar to RF equivalents, our phase-matched terahertz transmission lines confine the terahertz electric field as it builds up and guide it along with the optical pump, enabling longer interaction lengths beyond a few millimeters and supporting close to single-cycle terahertz pulses with amplitudes two orders of magnitude larger compared to previous demonstration. We

read out these terahertz electric fields all-optically, and demonstrate a detection close to shot-noise limit with a dynamic range exceeding 50 dB in intensity. Our phase-matching at telecommunication wavelengths is particularly significant, as there are no alternative bulk generation crystals that are phase matched at 1550 nm other than organic crystals[77]. Most generation crystals only work at other wavelengths, such as 780 nm[78] or 1050 nm[79]. This advantage opens doors for a wide variety of optical communication infrastructure to be implemented in our system. As an example, we use a fiber-based EO-amplitude modulator to control the amplitude of the terahertz emission. This not only provides future opportunities to integrate more functionalities on TFLN chips, but also showcases a path towards all-optical terahertz signal processing with gigahertz analog bandwidth. Combining the terahertz transmission lines with impedance-matching antennas for off-chip emission, additionally addresses free-space terahertz applications. Furthermore, we demonstrate the first proof-of-principle photonics-integrated terahertz strip-line cavities. We experimentally verify that efficient generation of specific modes is feasible by choosing the strip line length and placing a terahertz antenna at specific locations. Such a toolkit could provide a versatile way for tailoring terahertz emission for a specific selection of frequencies[80]. Finally, our system offers unique control over the terahertz radiative loss in the transmission lines through tweaking its dimensions and lower terahertz material loss because of thin layer of LN, providing access to a low-loss regime required for building more complex terahertz circuits. Our concept can be applied to continuous-wave applications requiring narrow linewidth terahertz signals by replacing the input pulse with two continuous-wave tones separated by a desired terahertz frequency[81]. Such all-optical generation of narrow-band terahertz tones can complement current all-electronic methods such as multiplier chains or quantum cascade lasers (QCL)[82]. This is beneficial in the context of modulators operating at terahertz frequencies for optical fiber communications by reducing the reliance on external instrumentation[83,84] with significant ohmic loss. Additionally, local oscillators in heterodyne terahertz spectroscopy would benefit from all-optical narrow-band terahertz tones without requiring cryogenic cooling, as seen with QCLs. Their capability to generate terahertz radiation above 1 THz could allow measuring hyperfine transitions[85] e.g., of [CII] at 158 $\mu$m[86], [NII] at 122 $\mu$m and 205 $\mu$m, Al above 2 THz. More importantly, their compatibility with in-situ tuning of the optical signals could provide the fine- and broad tuning needed for high-resolution planetary heterodyne spectroscopy.

Second, we find that the concept of phase matching using transmission lines supports efficient terahertz detection through electro-optic sampling on the same platform. We achieve 60 dB dynamic range and a 3 THz bandwidth with 100 ms integration time per point. Providing sub-wavelength confinement of terahertz waves, our phase-matched strip line cavities also facilitate quantum electrodynamics experiments which rely on sub-cycle field metrology[87,88]. Vacuum fields confined to these structures are estimated to be three orders of magnitude stronger than those in free space experiments[89], simplifying their direct detection by routing optical waveguides into these cavities. Finally, the combination of strongly enhanced local terahertz field inside the transmission lines, terahertz dispersion engineering (by means of designing the transmission line geometry) and extremely high nonlinear coefficient of TFLN ($d_{33}$ = 5870 pm/V[90] at terahertz wavelengths) can enable all-terahertz mixing (similar to all-optical mixing). These, combined with the mature wafer-scale fabrication of TFLN[91] allows for integration of numerous photonic components such as femtosecond pulse sources[92,93], electro-optic frequency combs[94], or self injection-locked laser sources with our terahertz circuits[95]. We anticipate that the design guidelines we propose will become crucial in future terahertz applications both on- and off-chip, such as free-space communications, terahertz-speed computing, data interconnects, ranging and metrology.

## Methods

### Fabrication

The chips are fabricated on 600 nm of X-cut lithium niobate, bonded to 4700 nm of thermally grown oxide on an $500\,\mu m$-thick double-side polished high-resistivity silicon substrate. The waveguides are patterned using electron-beam lithography (Eliionix ELS-HS50) and Ma-N resist. These waveguides are then etched into the LN layer using $Ar^+$ ions, followed by annealing in an $O_2$ environment to recover implantation damages and improve the absorption loss of the platform[96]. Subsequently, the devices are clad with 800 nm of Inductively Coupled Plasma - Chemical Vapor Deposition (Oxford Cobra), followed by another annealing. The electrodes are defined using a self-aligned process, including patterning with PMMA resist, dry etching, and lift-off using the same resist. The electrodes are deposited using electron-beam evaporation (Denton) with 15 nm of Ti and 300 nm of Au.

### Optical setup

The mode-locked laser used for THz generation and detection is C-Fiber 780 from Menlo Systems. it provides 100 MHz repetition rate with average power values up to 500 mW and pulse duration in the range of 60 fs. When TFLN is used as the terahertz source (Figs. 2, 3, and 4), the 1560 nm output is used as the pump for THz generation and the 780 nm output is used as the probe for electro-optic sampling. Before coupling the pump laser into fiber, a silicon prism pair (SIFZPRISM25.4-BREW from Crystran) is used in a double pass configuration to provide tunability of group velocity dispersion and the spectrum is filtered using a band-pass filter with FWHM of 12 nm at 1530 nm (FBH1530-12 from Thorlabs). To precompensate the total dispersion from the 11 m of single mode fiber inside the setup, we use dispersion compensating fiber (2x PMDCFA5 from Thorlabs). Modulation of the pump is done using the component MXAN-LN-10-PD-P-P-FA-FA from IX-Blue, which can support up to 10 GHz. Losses from coupling to fiber and form the intensity modulator are compensated using the optical amplifier EDFA100S from Thorlabs which operates at maximum pump current. After amplification, the high peak power in fiber triggers spectral broadening (from 12 nm to 50 nm) and Raman scattering leads to shift in spectrum toward longer wavelength (Fig. 5b in Supplementary Information Sec. 1C). The broad spectrum leads to efficient optical rectification inside $LiNbO_3$. The polarization is adjusted before coupling to the chip using a polarization controller to couple into the TE mode of the x-cut $LiNbO_3$ waveguide and exploit $d_{33}$ coefficient. Finally, a 1 m bare lensed fiber with $5\,\mu m$ minimum spot-size is used for edge coupling and the coupling efficiency from fiber to the chip reaches 13%. For measuring the signal using the conventional electro-optic sampling technique, a ZnTe and GaP crystals are used. The pulse length of the 780 nm probe is 58 fs. Parabolic mirrors with 50 mm reflective focal length are used to collect and focus the THz radiation. The probe signal is measured using a balanced photodetector (PDB465A from Thorlabs) UHFLI 600 MHz Lock-in Amplifier with time constants between 300 ms and 1 s. The delay stage scans the THz signal by moving the probe pulse in time with 25 fs steps. To detect terahertz using TFLN (Fig. 5), we use the 780 nm as the pump for a commercial GaAs PCA (iPCA-21-05-1000-800-h from Batop GmbH) and the photodetector Nirvana 2017 for reading out the 1560 nm probe from the chip.

## Data availability

The data generated in this study have been deposited in the Zenodo database under (https://doi.org/10.5281/zenodo.15316587).

## Code availability

The code used to plot the data within this paper is available in the Zenodo database under (https://doi.org/10.5281/zenodo.15316587).

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

## Acknowledgements

Y.L., A.G., and I.C.B.C. acknowledge funding from the European Union's Horizon Europe research and innovation program under project MIR-AQLS with grant agreement No 101070700. S.R., A.T., X.C., and I.C.B.C. acknowledge funding from the Swiss National Science Foundation under PRIMA Grant No. 201547. S.R. acknowledges financial support from the Hans Eggenberger foundation (independent research grant 2022) and from the Swiss National Science Foundation (Post-doc.Mobility, grant number 214483). A.S.-A. and M.L. acknowledge funding from Defense Advanced Research Projects Agency (HR0011-20-C-0137) and funding from National Science Foundation (NSF) (ECCS-2407727). L. M. acknowledges Capes-Fulbright and Behring foundation fellowships. The fabrication of these chips was performed in part at the Center for Nanoscale Systems (CNS), a member of the National Nano-technology Coordinated Infrastructure Network (NNCI), which is supported by the National Science Foundation under NSF Award no. 1541959. The views, opinions and/or findings expressed are those of the author and should not be interpreted as representing the official views or policies of the Department of Defense or the U.S. Government.

## Author contributions

I.C.B.C., Y.L., and A.S.-A. conceptualized the project. Y.L. built the terahertz-optical setup and carried out the measurements. A.T. built the detection setup. S.R. assisted with building the optical setup. Y.L. and A.G. performed the CST simulations. X.C. assisted with the detector measurements. S.R. assisted with the CST simulations of the antenna design. A.S.-A., S.R., and L.M. fabricated the devices. A.G. derived the theoretical description of the terahertz emission. Y.L., A.S.-A., A.G., and I.C.B.C. wrote the manuscript with feedback from all authors. I.C.B.C. and M.L. supervised this work.

## Competing interests

Y.L., A.S.-A., A.T., M.L., and I.C.B.C. are inventors on U.S. patent application US18/213,691. The remaining authors declare no competing interests.
