## [Transparent Peer Review file · Nature Communications]

Photonics-integrated terahertz transmission lines

Corresponding Author: Professor Ileana-Cristina Benea-Chelmus

Version 0:

Reviewer comments:

Reviewer #1

(Remarks to the Author)

This manuscript presents integrated phase-matched THz transmission lines with thin-film lithium niobate (TFLN), demonstrating broadband THz generation through optical rectification when pumped with femtosecond optical pulses. In comparison to the authors' previous work (ref 33), which demonstrated THz waveform synthesis in an integrated TFLN platform, the authors demonstrate significantly higher generated THz fields and broader radiation bandwidth with the new phase-matched THz transmission lines.

I have the following suggestions for the authors:

1. When comparing results with the state-of-the-art, it is unclear what is meant by "state-of-the-art." It appears that the authors are referring to their previous work (ref 33) as the state-of-the-art, but it would be more appropriate to compare their results to other integrated THz sources pumped by external femtosecond lasers. There is a significant body of work on integrated THz sources. Especially, there are many prior demonstrations where the generated THz signal from on-chip THz sources propagates through on-chip transmission lines and is detected by on-chip THz detectors. I suggest that the authors cite and benchmark their results against such work. Examples include (though there are many more relevant publications):

- Yoshioka, et al., 2024. On-chip transfer of ultrashort graphene plasmon wave packets using terahertz electronics. *Nature Electronics*, pp.1-8.
- Potts, et al., 2023. On-chip time-domain terahertz spectroscopy of superconducting films below the diffraction limit. *Nano Letters*, 23(9), pp.3835-3841.

2. In comparing their results with the state-of-the-art, the authors should account for the optical pulse width used to pump the on-chip THz sources. For example, when comparing with ref 33, it is difficult to determine how much of the improvement is due to better phase matching in the new design versus the use of narrower pump pulses. In ref 33, the femtosecond optical pump is coupled through a grating coupler and distributed to multiple on-chip devices, which could broaden the optical pulse, reducing the efficiency of optical rectification and limiting the bandwidth of the generated THz signal. However, in this manuscript, the experimental setup uses special precautions and dispersion compensation to generate 60 fs optical pump pulses. Therefore, it is not an entirely fair comparison. For instance, if the authors were to excite their device with even shorter optical pulses broader THz bandwidth and stronger fields could be achieved due to enhanced optical rectification, rather than improved phase matching.

3. The authors note limitations of THz generation using non-collinear phase matching and tilted pulse front techniques, mentioning the non-Gaussian THz beam shape as a shortcoming. However, the device demonstrated in this manuscript faces the same issue. The authors should clarify this point to ensure consistency.

4. Reporting THz field is not an accurate way to assess the efficiency of THz generation, especially when off-chip free-space components are used for field measurements. I suggest that the authors report the generated THz power, which would allow them to account for power directed toward the dielectric and not be limited by the numerical aperture of their detection system.

5. The authors should include specifications of their time-domain measurements, such as the number of time-domain traces used to produce the results in Fig. 2, and the total duration of the measurements.

6. The authors state, "we demonstrate hybrid integration of THz transmission lines with photonic integrated circuits." However, this work demonstrates hybrid integration of THz transmission lines with TFLN waveguides, while other photonic components are off-chip. I suggest the authors use more precise terminology.

Reviewer #2

(Remarks to the Author)

This paper presents recent advances in photonics-integrated terahertz transmission lines. Here, a thin-film lithium niobate (TFLN) platform is used to fabricate active terahertz waveguides where gain is realized by optical rectification of telecom wavelength pulses. In this way, a frequency bandwidth between 200 GHz and 3.5 THz is covered. The novelty here is the integration of such a rectification device into a THz waveguide where the phase matching to the optical mode is carefully engineered.

In general, all results of the manuscript are supported by an analytical description and/or field simulation combined with measurements. The only thing the authors should do is to use a calibrated terahertz powermeter to measure the total radiated terahertz power. This will validate the results of the partly very indirect calculations.

This work may have an impact in the future because it is a very versatile platform. However, the authors must work on the state of the art. In my opinion, the biggest competitors in this field are not free-space optical rectification devices, but photoconductive antennas and photodiodes. With these, also telecom compatible devices (also on the receiving side), peak dynamic ranges exceeding 130 dB and bandwidths exceeding 10 THz have already been demonstrated. An advantage of the proposed method is the relatively flat emission spectrum, but how does the emitted power (in mW) compare and how does the efficiency compare? What can be done with these devices that cannot be done with the established PCA/diode emitters? Is there a way to integrate a detector in a similar way?

<https://doi.org/10.1364/OE.454447>

<https://doi.org/10.1038/s41598-023-40634-3>

In addition, I have the following questions/comments.

1. Figure 1. A) It is not clear what can be seen on the picture of the fabricated photonics-integrated terahertz transmission lines. Also, there are many examples of "photonic" structures in the terahertz range, such as photonic crystals, dielectric waveguides, etc.
2. Figure 1. C) Why $n(f_{\text{THz}})$ and not n_{TL} ? This needs to be consistent throughout the manuscript.
3. Fig 1. D) It would be interesting to see the full scale of this graph.
4. Line 79: lowest frequency is missing
5. Is it possible to integrate an optical source with the emitter?
6. Line 116: Please state the bandwidth of the optical laser also as a frequency and not only as 75 nm.
7. Why do you distinguish between optical wave and terahertz fields in line 123?
8. What does the "*" in eq. 3 mean for g^*_{THz} ? Complex conjugated? But these are not complex functions in the time domain, are they?
9. Figure caption 2: Is the length of the transmission line really 32 mm? I think this is a typo.
10. Lines 162-164: What window function was used to compute the FFT? And please add some details about the data acquisition: Step mode, integration time, and total acquisition time. Especially the acquisition time is important to contextualize the obtained DR.
11. Line 183: Pure amplitude modulation is of course not a terahertz communication will be realized. There are better modulation schemes.
12. Line 196: Why is the 120 μm trace not shown anywhere? Only 2 mm, 100 μm and 200 μm . Why don't you show traces for all lengths?
13. Lines 219-220: Two times odd modes for strong and weak excitation. One is wrong.
14. Figure 4. C) The 200 μm transmission line shows a double pulse with the same amplitude. What I would expect from a transmission line filter is a pulse train with increasing amplitude (depending on the Q-factor).
15. Why is your Q-factor so low? There must be a huge impedance mismatch at both ends of the line. Are the losses underestimated?
16. Line 280: The term "terahertz speed modulators" seems a bit strange to me. Maybe it is better to call them "modulators operating at terahertz frequencies".
17. Line 307: The waveguides were patterned using electron beam lithography: Does the platform also allow conventional lithography?
18. Why is the 200-fs probe pulse so long? Usually you want to use the shorter pulse as the probe pulse.
19. Why is the refractive index measured at about 1500 nm and not at 1550 nm?
20. Supplement Figure 5 c): Black and brown are not good choices for coloring.

Version 1:

Reviewer comments:

Reviewer #1

(Remarks to the Author)

The additional analysis and revisions have indeed improved the manuscript.

As the second referee also noted, the main competition to the demonstrated THz source and detector are photoconductive antennas (PCAs). Numerous prior studies have demonstrated THz signal generation and detection using on-chip PCA sources and detectors integrated with THz transmission lines (as referenced in my previous comments). These on-chip PCAs achieve significantly higher terahertz powers (on the order of milliwatts) and higher SNRs compared to the TFLN devices presented in this manuscript, which are expected to generate only 0.2 nW of power. Unfortunately, the authors dismiss PCAs by stating that they require a free-space optical pump, while their TFLN devices also rely on a free-space optical pump. The only difference is that the free-space pump is coupled into an optical waveguide in the TFLN devices to satisfy a phase-matching condition—a requirement that does not apply to PCAs. Therefore, the manuscript does not convincingly demonstrate how the TFLN devices advance the current state of the art.

Reviewer #2

(Remarks to the Author)

I would like to thank the authors for their detailed response. All of my critical comments have been addressed thoroughly and appropriately. I therefore recommend accepting the manuscript in its current form.

I do have one minor remark: As the authors mention, some of the best PCAs come from HHI. A comparison with these antennas would therefore be worthwhile. I am, of course, aware that HHI technology has been developed and refined over many years. Nevertheless, Batop antennas should not be considered a reference. However, since the focus here is more on replacing power measurements, the approach taken is acceptable

Version 2:

Reviewer comments:

Reviewer #1

(Remarks to the Author)

Response to referees

We would like to thank all reviewers for their constructive assessment of our work and their inspiring suggestions. In detail, we would like to thank the referee 1 for highlighting the significant performance we achieve, and referee 2 for underlining the versatility of our platform and its potential future impact.

Given the suggestions and comments we received in the first round of reviews, we have decided to undertake the following **major revisions** to the initially submitted version of the manuscript and supplementary information:

1. In the introduction, we complemented our discussion of bulk terahertz generation with a discussion on integrated semiconductor technologies expanding the scope of our state of the art and underlining the advantages and disadvantages of each platform (Comment 2, referee 1, Comment 3 referee 2).
2. We compare our TFLN emitters to a commercially available PCA in supplementary Sec. 6.B "Benchmarking of our TFLN emitters to commercial photoconductive antenna" (Comment 6, referee 1, Comment 3, referee 2)
3. The originally submitted manuscript only focused on generation. We now add detection of broadband terahertz pulses via on-chip electro-optic sampling which further strengthens our claim about the advantage of long phased-matched transmission lines. We have therefore added the section "Broadband terahertz detection" to the main text, the used optical setup to the supplementary Sec. 1.D "Experimental setup for terahertz detection", the details of the used components to the methods section, and the theoretical derivation of detection in supplementary Sec. 4.A.
We further benchmark our detector to a commercially available PCA in the supplementary Sec. 6.C "Benchmarking of our TFLN detectors to commercial photoconductive antenna". Finally, we study the effect of the transmission line length for detection as well, which we added to the supplementary Sec. 5.B "Effect of transmission line length on terahertz detection" (Comment 5, referee 2).
4. To compare the different integrated platforms in a quantitative way, we added the supplementary Sec 6.D "Overview of key metrics for integrated hybrid terahertz devices" (Comment 2, referee 1 and Comment 3, referee 2). As part of this analysis, we highlight the importance of high-power handling for broadband generation of terahertz pulses (which typically requires short femtosecond pulses that carry high peak powers). Large-bandgap materials such as lithium niobate are superior to low-bandgap materials which suffer from strong two-photon absorption and are often used for photoconductive devices. This is particularly relevant on-chip, where light is confined to waveguides and can reach high intensities already at relatively modest average powers. This is now found in supplementary Sec. 6.A "Power handling of common terahertz materials" (Comment 4, Referee 2)

Furthermore, we made the following minor revisions to the manuscript:

1. To quantify how the pulse length influences the bandwidth of the generated THz radiation, we provide a mathematical derivation that links these two physical properties in supplementary Sec. 2.B. "Effect of pulse length on THz bandwidth" (Comment 3, referee 1) and added the required maximal pulse length for each THz frequency to Fig. 1 d (right y-axis). Together with the discussion on phase matching, this should provide a clearer summary over the experimental constraints that need to be fulfilled in order to achieve a certain bandwidth. (Comment 3, referee 1)
2. We enlarged the scale of the left y-axis in Fig. 1 d of the main text to show the coherence length (Comment 9, referee 2)
3. We estimated the generated power and generation efficiency and added it to supplementary Sec. 3.A "Estimation of collection and detection efficiencies" (Comment 6, referee 1, and Comment 2, referee 2)
4. Fig. 1 in the main text has been modified: SEM pictures of the device have been added to the device. Caption of Fig. 1 has been modified to describe the new figure (Comment 6, referee 2)
5. Sentence "In the context of sensing and spectroscopy, they offer confinement of terahertz radiation below the diffraction limit, increasing the spatial overlap with the sensed medium, boosting sensitivity [1, 2]" has been added to the introduction (Comment 2, referee 1)
6. Sentence "By providing short pump pulses to our devices, the spatial extension of the pulse becomes shorter than the transmission line and phase matching with the terahertz pulse is required for high efficiency" has been added to describe Fig. 1 (Comment 3, referee 1)
7. We study the effect of adding a silicon lens to the detector's performance in supplementary Sec. 5.A "Effect of silicon lens on terahertz detection"
8. Sentence "a field amplitude higher than previous work in TFLN by two orders of magnitude (Fig. **A2** b, middle panel) [3] has been added to the introduction" (Comment 2, referee 1)
9. Transmission line length has been corrected to 2 mm in the caption of Fig. 2 (Comment 15, referee 2)
10. We added the details on acquiring the signal "The measurement represents a single time trace, acquired with a step size of 25 fs and an integration time of 100 ms per time point." (Comment 7, referee 1 and Comment 16, referee 2)
11. We added the following sentence to the section Electro-optic control of the terahertz field amplitude "Amplitude modulation is one of the cornerstones of current communication schemes and precise control over the

amplitude of the generated terahertz fields is essential for future generations of terahertz devices” (Comment 17, referee 2)

12. We improved the explanation of section Terahertz cavities from coplanar transmission lines by adding ”The remaining terahertz field continues to build up until it reaches the end of the transmission line and reflects back. The optical pump leaves the structure, and the remaining terahertz radiation experiences net loss during propagation. Once the terahertz pulse reaches the antenna again, it is coupled out with a group delay of $t_g = \frac{n_g L_{\text{TFL}}}{c}$ with respect to the initial terahertz pulse.” (Comment 20, referee 2)
13. The beginning of discussion has been changed ”In summary, we demonstrate that the hybrid integration of terahertz transmission lines with TFLN waveguides and dipole antennas enables compact and efficient solutions for the generation and detection of broadband terahertz radiation. First, we demonstrate broadband and spectrally flat terahertz generation up to 3.5 THz, spanning four octaves. The pump pulses are provided by an off-chip commercial mode-locked laser connected to a telecom fiber system and the detection is done with conventional electro-optic sampling. We achieve this bandwidth by overcoming the significant limitation of phase mismatch between the optical and terahertz signals, which previously caused their early walk-off in bulk”
14. The pulse length of the probe in the section optical setup has been corrected to 58 fs (Comment 24, referee 2)

We hope that altogether, the demonstrated generation of broadband terahertz pulses via optical rectification of femtosecond pulses, detection of broadband pulses via on-chip electro-optic sampling, the realization of photonics-integrated terahertz cavities and the electro-optic control of terahertz pulses convincingly showcase the potential of terahertz transmission lines implemented in thin film lithium niobate for integrated terahertz devices.

Questions are shown in blue, answers are shown in black and changes to the manuscript are shown in red.

1 Reviewer #1

1. This manuscript presents integrated phase-matched THz transmission lines with thin-film lithium niobate (TFLN), demonstrating broadband THz generation through optical rectification when pumped with femtosecond optical pulses. In comparison to the authors' previous work (ref 33), which demonstrated THz waveform synthesis in an integrated TFLN platform, the authors demonstrate significantly higher generated THz fields and broader radiation bandwidth with the new phase-matched THz transmission lines. I have the following suggestions for the authors:

We thank the referee for the summary and for highlighting the improvements from previous work.

2. When comparing results with the state-of-the-art, it is unclear what is meant by "state-of-the-art." It appears that the authors are referring to their previous work (ref 33) as the state-of-the-art, but it would be more appropriate to compare their results to other integrated THz sources pumped by external femtosecond lasers. There is a significant body of work on integrated THz sources. Especially, there are many prior demonstrations where the generated THz signal from on-chip THz sources propagates through on-chip transmission lines and is detected by on-chip THz detectors. I suggest that the authors cite and benchmark their results against such work. Examples include (though there are many more relevant publications):
 - Yoshioka, et al., 2024. On-chip transfer of ultrashort graphene plasmon wave packets using terahertz electronics. *Nature Electronics*, pp.1-8.
 - Potts, et al., 2023. On-chip time-domain terahertz spectroscopy of superconducting films below the diffraction limit. *Nano Letters*, 23(9), pp.3835-3841.

We thank the referee for the remarks and acknowledge a lack of precision on our side. While many works, including those cited by the referee, have combined terahertz circuits with optical pumping, the optical pumps were often delivered from free-space, and not through a photonic circuit. The aim of our work is to close this gap, and showcase how the design freedom offered by integrated photonic platforms (PICs) can be beneficial to terahertz science and technology. In our paper, we focus on engineering the waveguide geometric dispersion to overcome the large index mismatch of bulk lithium niobate and enable collinear phase matching between optical and THz modes leading to broadband THz generation.

On the terahertz side, prior demonstrations of on-chip THz circuits have inspired the design of the terahertz components we integrated with our photonic circuit.

We agree with the referee that this needs to be covered in the introduction and take this into account in the list of changes below.

Apart from these studies, there are three main integrated photonic plat-

forms that are currently the subject of current research for realizing integrated terahertz photonic devices: III-V integrated photonic circuits, thin film lithium niobate and hybrid silicon-organic. To give readers a better overview of these technologies, we extended our comparison by providing a table in the supplementary material that compares the reported performances in their compatibility with CW/pulsed, power handling, terahertz and optical capabilities, bandwidth, generation efficiency and transparency window. We show the table **A1** below as well:

	LiNbO ₃	Silicon Organic	GaAs/InP
Scheme			Optical pumping	CW/pulsed	CW/pulsed	CW
Demonstrated on-chip powers (pulsed)	more than 360 mW [4]	few mW [5]	few tens of μ W [6, 7]
THz capabilities	generation[*], detection[*][8], waveguide[*], cavity [*]	detection [9], waveguide [9]	generation, detection, waveguide [1], cavity [10]
Optical capabilities	modulators, high Q cavities, filter, isolators [11]	modulators, high Q cavities, filters, detectors [12]	modulators [13], lasers [14, 15], detectors [16, 17]
Bandwidth	2.25 THz (10 dB), 3.5 THz (noise floor) [*]	1 THz (10 dB), 2 THz (noise floor) [9]	0.5 THz (10 dB), 4.5 THz (noise floor) [18]
Generation efficiency	10^{-7} [*]	generation not demonstrated	10^{-5} [19]
Optical transparency	350 nm to 5 μ m	above 1107 nm for Silicon	above 870 nm for GaAs above 925 nm for InP

Table **A1**: Integrated hybrid optical-terahertz platforms. CW: continuous wave, [*]: this work

Changes to the manuscript: To address this comment from the referee, we took the following steps:

- (a) We significantly expanded our state-of-the-art to include other integrated photonic platforms, including III-V and hybrid silicon-organic, lines 39-45 and lines 53-77.
- (b) We added a discussion about prior work in integrated terahertz components (transmission lines, split ring resonators) and their applica-

tions in spectroscopy to the introduction, lines 80-89.

(c) We provide a quantitative comparison of the different integrated platforms in supplementary Sec. 6 D "Overview of key metrics and properties for integrated hybrid terahertz devices".

3. In comparing their results with the state-of-the-art, the authors should account for the optical pulse width used to pump the on-chip THz sources. For example, when comparing with ref 33, it is difficult to determine how much of the improvement is due to better phase matching in the new design versus the use of narrower pump pulses.

We thank the referee for pointing out this lack of clarity from our side. In line with the equation **A1** in the main text, we fully agree that high conversion efficiency and large bandwidth require both phase matching and pumping with short pulses since

$$\tilde{E}_{\text{THz}}(l_{\text{TL}}, \omega_{\text{THz}}) = \frac{i\pi\chi^{(2)}E_0^2\omega_{\text{THz}}^2\tau^2l_{\text{TL}}\Gamma_{\text{overlap}}}{4c_0n_{\text{TL}}(\omega_{\text{THz}})\sinh(\pi\omega_{\text{THz}}\tau/2)}G_{\text{TL}}(\omega_{\text{THz}}) \cdot e^{-i\frac{\omega_{\text{THz}}l_{\text{TL}}}{c_0}n_{\text{TL}}(\omega_{\text{THz}})} \quad (\text{A1})$$

which can be reformulated assuming a constant optical pulse energy J_{opt} and perfect phase matching to describe the dependency on pulse length

$$E_{\text{THz}} \propto J_{\text{opt}} \frac{\omega_{\text{THz}}^2 \cdot \tau \cdot l_{\text{TL}}}{\sinh(\pi\omega_{\text{THz}}\frac{\tau}{2})} \quad (\text{A2})$$

From this equation, we see that both the pulse length and the length of the transmission line play a role. We can further simplify the sinh for the case that the supported bandwidth of the pump pulse is larger than the one supported by phase matching or THz losses $T_{\text{THz}} \gg \tau$ to:

$$E_{\text{THz}} \propto J_{\text{opt}} \frac{\omega_{\text{THz}}^2 \cdot \tau \cdot l_{\text{TL}}}{\sinh(\pi\omega_{\text{THz}}\frac{\tau}{2})} \approx 2J_{\text{opt}} \frac{\omega_{\text{THz}} \cdot l_{\text{TL}}}{\pi} \quad (\text{A3})$$

which shows that the amplitude of the terahertz electric field obtained by optical rectification increases with terahertz frequency linearly, if not otherwise constrained by the pulse length of the pump pulses or by phase matching. This is an important distinction compared to photoconductive devices.

Therefore, as the referee correctly points out, we optimize *both* in our device, impacting the bandwidth and amplitude of our emitter. To answer the referee's question in quantitative way, we show how the terahertz power is affected for various pulse lengths (Fig. **A1 a**). From this scaling, we can determine the required pulse length for efficiently generating at a given terahertz frequency. Next, we take the frequency at which the power drops by factor 2 and find the corresponding pulse length. We thereby quantify

Figure A1: **a** Power scaling from equation A3 assuming constant pump energy for different pulse lengths. **b** Power values normalized to the case of shortest pulse length $\tau = 1$ fs. Dashed highlights the 3-dB drop in normalized power

that the pump pulse length must be below 120 fs to obtain sufficient power values at 3 THz.

Changes to the manuscript: In our original version of the manuscript, we visually depict the longer pulse length of Ref. [3] by making the pulse at the left panel of in Fig. 1 b longer - we now see that this may easily go unnoticed. To improve the clarity we made the following changes:

- (a) we now specifically mention the maximum required pulse length to achieve a 3-dB bandwidth for all THz frequencies on the right y-axis Fig. 1 d. This is in line with the referee's comment on importance of phase matching and pulse length, and aims to summarize both in a single graph.
 - (b) We explain how we quantify this plot in supplementary Sec. 2B "Effect of pulse length on the terahertz bandwidth".
4. In ref 33, the femtosecond optical pump is coupled through a grating coupler and distributed to multiple on-chip devices, which could broaden the optical pulse, reducing the efficiency of optical rectification and limiting the bandwidth of the generated THz signal. However, in this manuscript, the experimental setup uses special precautions and dispersion compensation to generate 60 fs optical pump pulses. Therefore, it is not an entirely fair comparison. For instance, if the authors were to excite their device with even shorter optical pulses broader THz bandwidth and stronger fields could be achieved due to enhanced optical rectification, rather than improved phase matching.

We thank the referee for the valid question. The referee points out that Ref. [3] uses grating couplers which result in longer pulses (about 500 fs), and that broader spectra would be expected if the pulse was reduced to the value we achieve in our work (60 fs). Fig. 4 of our main manuscript

captures precisely the point made by the referee. These antennas have short interaction lengths of $l_{TL} = 100, 200 \mu\text{m}$ (as in Ref. [3]) but they are pumped with 60 fs pulses. We see that the resulting terahertz fields are broadband up to 3 THz, as anticipated by the referee. However, we also find that for these high frequencies, the transmission line acts as a resonator, because

- (a) its length is larger than half a wavelength
- (b) the open-ended transmission line has a high reflectivity, quantified through S_{11}

Consequently, at frequencies above the fundamental frequency of the transmission line ($f_0 = \frac{c_0}{2n_{TL}L_{TL}}$) the spectrum is shaped, following a pattern of modes that are efficiently out-coupled (odd) and those that are not (even). For frequencies below the fundamental $f < f_0$, we're back to exactly the scenario of Ref. [3]. In this frequency range, the length of the transmission line is sub-wavelength. We tried to draw attention to this regime also in the spectra of Fig. 4d by labeling it as "subwavelength".

We compare in Fig. **A2** the results presented in Fig. 4 of the main manuscript and the ones of Ref. [3]. The 60 fs pulse length we achieve in our case clearly offer broader spectra compared to the 500 fs used in the earlier work [3].

Changes to the manuscript: We emphasize the limited pulse length of Ref. [3] in the text, lines 120-122.

5. The authors note limitations of THz generation using non-collinear phase matching and tilted pulse front techniques, mentioning the non-Gaussian THz beam shape as a shortcoming. However, the device demonstrated in this manuscript faces the same issue. The authors should clarify this point to ensure consistency

We agree with the referee that the farfield of our TFLN emitter may deviate from a Gaussian beam, since we exploit the higher order modes of the dipolar antenna.

Changes to the manuscript: We now point out this aspect in the main text, lines 206-208.

6. Reporting THz field is not an accurate way to assess the efficiency of THz generation, especially when off-chip free-space components are used for field measurements. I suggest that the authors report the generated THz power, which would allow them to account for power directed toward the dielectric and not be limited by the numerical aperture of their detection system.

We agree with the referee that the performance is highly underestimated due to the limitations in collecting and focusing the radiation. Nevertheless, we decided to adopt this "conservative" and simple approach for the

Figure A2: Effect of the pulse length on terahertz emission. **a** used device for comparing the effect of the pulse length. In our case, edge coupling is used to couple the entire bandwidth into the chip. We measure 60 fs and the previous work [3] reports a pulse length in the range of 500 fs. **b** Measured time domain signals. The measurement from the current work corresponds to Fig4. in the main text. We obtain the signal with 25 fs step size, 100 ms integration time. The measurement from [3] is multiplied by 10 for better visibility. **c** Fourier transform of **b** highlights the importance of the pulse length even when phase matching is fulfilled.

following reason: we believe that the as-measured terahertz electric field inside the ZnTe detection crystal is the most transparent way to showcase the performance of our chip since it does not correct for any terahertz loss that may easily be overseen or misunderstood by readers, for example at the crystal’s surface or at the collecting parabolic mirror.

Referee 2 has suggested measuring the power directly with a THz power meter and we responded as follows: We are in possession of the power meter ”Erickson PM5B” from Virginia Diodes Inc, which can measure average power values down to micro watt but it is meant to operate at sub-THz frequencies in the CW regime. We estimated the THz energy from our measured field values in Supplementary Sec. 3.A ”Estimation of collection and detection efficiencies” and find $J_{\text{THz}} = 2.3 \cdot 10^{-18}$ J which corresponds to an average power of $P_{\text{THz}} = J_{\text{THz}} \cdot f_{\text{rep}} = 0.2$ nW. Unfortunately, we

Figure A3: Comparison of commercial PCA and our device in Fig. 2 in the main text **a** Simplified sketch of the optical setup for pumping the our lithium niobate terahertz emitter. A hyperhemispherical silicon lens is mounted on the backside of the chip. **b** same setup is used to pump a commercial photoconductive antenna which is biased with 10 V DC voltage. **c** Measured optical spectrum after the EDFA for 12 mW and 60 mW average power where the power is changed by changing the EDFA gain. High optical power leads to spectral broadening inside the fiber. **d** Autocorrelation measurements give 222 fs, and 63 fs for 12 mW and 60 mW accordingly. **e** measurement results from EO sampling show that our emitter can surpass commercial PCAs at higher terahertz frequencies and does not suffer from power limitation. Measurement conditions are provided in Supplementary table 8.

are not in possession of power meters that can measure THz power values in the range of nW. Adjustments, such as higher coupling to the chip, higher optical power, longer transmission line, improved antenna design, and proper lens design to out-couple the radiation into free-space can potentially achieve μW values since the power scales quadratically with field amplitude.

Given these practical constraints, we concluded that the best approach is to compare our chip to a commercial antenna-coupled photoconductive switch pumped at 1550 nm.

The device we compare to is "bPCA-180-05-10-1550-h" from Batop GmbH. Since we are targeting integrated devices, we chose this commercial device that requires a low bias voltage of 10 V and contains a single antenna with similar dimensions to our emitter. The PCA has a silicon lens on the back side. We have attached a silicon lens to the back side of our TFLN chip too, since both the substrate and the lens are made of high resistivity silicon. To ensure a fair comparison, both were measured under the exact same conditions illustrated in Fig. **A3** c and d. We compare our device from Fig. 2 to the PCA under two measurement conditions: at 12 mW optical pump power (inside the optical fiber) to prevent saturation of the PCA, and at 60 mW inside the optical fiber where we achieve spectral broadening inside the fiber. As shown in Fig. **A3** e (left panel), at low optical powers (corresponding to pump pulses of 222 fs) the PCA performs better at frequencies below 1 THz (by a factor of approx. 20 dB in power). At frequencies above 1 THz, the two are approximately similar in performance. We note that the PCA spectrum we measure agrees well with the one reported in the datasheet [20]. At 60 mW optical power (corresponding to the optical power used in all measurements of the main manuscript), our device exceeds the PCA at frequencies larger than 500 GHz by 10-20 dB. We note that we mounted the Si-lens on our TFLN chip manually which may lead to non-ideal alignment. For both measurements, we use 1 MHz modulation frequency, 100 ms integration time, and 25 fs step size.

Despite these efforts, we note that the two devices are still quite different in nature, since the PCA does not contain any integrated photonic circuit, limiting miniaturization.

Changes to the manuscript: To ease the comparison between our chips and other solutions we made the following changes:

- (a) We added the power estimation to supplementary Sec. 3A "Estimation of collection and detection efficiencies".
- (b) We added the supplementary Sec. 6B "Benchmarking of our TFLN emitters to commercial photoconductive antenna".

7. The authors should include specifications of their time-domain measurements, such as the number of time-domain traces used to produce the

results in Fig. 2, and the total duration of the measurements.

We thank the referee for the important remark. The specifications are 25 fs step size of the delay line and integration time of 100 ms. The same settings are used for all measurements.

Changes to the manuscript: To address this comment, we now mention these experimental settings in the main text lines 216-218.

8. The authors state, "we demonstrate hybrid integration of THz transmission lines with photonic integrated circuits." However, this work demonstrates hybrid integration of THz transmission lines with TFLN waveguides, while other photonic components are off-chip. I suggest the authors use more precise terminology.

We thank the referee for the important remark. Indeed, the highlighted sentence is not accurate.

Changes to the manuscript: We clarify this part of the discussion as follows: "In summary, we demonstrate that the integration of terahertz transmission lines with TFLN waveguides and dipole antennas enables compact and efficient solutions for the generation and detection of broadband terahertz radiation. First, we demonstrate broadband and spectrally flat terahertz generation up to 3.5 THz, spanning four octaves. The pump pulses are provided by an off-chip commercial mode-locked laser connected to a telecom fiber system and the detection is done with conventional electro-optic sampling. We achieve this bandwidth by overcoming the significant limitation of phase mismatch between the optical and terahertz signals, which previously caused their early walk-off in bulk"

2 Reviewer #2

1. This paper presents recent advances in photonics-integrated terahertz transmission lines. Here, a thin-film lithium niobate (TFLN) platform is used to fabricate active terahertz waveguides where gain is realized by optical rectification of telecom wavelength pulses. In this way, a frequency bandwidth between 200 GHz and 3.5 THz is covered. The novelty here is the integration of such a rectification device into a THz waveguide where the phase matching to the optical mode is carefully engineered. In general, all results of the manuscript are supported by an analytical description and/or field simulation combined with measurements.

We thank the referee for the summary and for their positive remarks on the work.

2. The only thing the authors should do is to use a calibrated terahertz power meter to measure the total radiated terahertz power. This will validate the results of the partly very indirect calculations.

We thank the referee for their input. The previous referee has also required to report the terahertz power and we respond as follows: We are in possession of the power meter "Erickson PM5B" from Virginia Diodes Inc, which can measure average power values down to micro watt values but it is meant to operate at sub-THz frequencies in the CW regime.

We estimated the THz energy from our measured field values in Supplementary section 3 A. "Estimation of collection and detection efficiencies" to be $J_{\text{THz}} = 1.1 \cdot 10^{-18}$ J which corresponds to an average power of $P_{\text{THz}} = J_{\text{THz}} \cdot f_{\text{rep}} = 0.1$ nW. Unfortunately, we are not in possession of power meters that can measure THz power values in the range of nW. Adjustments, such as higher coupling to the chip, higher optical power, longer transmission line, improved antenna design, and proper lens design to out-couple the radiation into free-space can potentially achieve μW values since the power scales quadratically with field amplitude.

Changes to the manuscript: To ease the comparison between our chips and other solutions, we added the power estimation to supplementary Sec. 3A "Estimation of collection and detection efficiencies".

3. This work may have an impact in the future because it is a very versatile platform. However, the authors must work on the state of the art. In my opinion, the biggest competitors in this field are not free-space optical rectification devices, but photoconductive antennas and photodiodes. With these, also telecom compatible devices (also on the receiving side), peak dynamic ranges exceeding 130 dB and bandwidths exceeding 10 THz have already been demonstrated. An advantage of the proposed method is the relatively flat emission spectrum, but how does the emitted power (in mW) compare and how does the efficiency compare?

We thank the referee for the important remark. Indeed, we also believe that integrated photonic circuits bring along many functionalities in routing, multiplexing or switching that can not be easily achieved in bulk. Al-

though we believe that a comparison to bulk lithium niobate components is necessary in the introduction in order to appreciate the advantages of the material and disadvantages of bulk realizations that may be mitigated on-chip, we agree that a discussion on the state of the art of PCAs will give readers a more complete view over the advantages and disadvantages of each platform.

Perhaps the most fundamental distinction that can be made between generation via photoconduction in PCAs and optical rectification in TLFN is the scaling of these two processes with terahertz frequency: while optical rectification becomes increasingly efficient for higher terahertz frequencies (the generated terahertz electric field amplitude increases linearly with frequency as imposed by Eq. **A3**), the efficiency of photoconductors strongly decreases with terahertz frequency. For example, in papers mentioned by the referee, the -20 dB cut-off is typically around 500 GHz-1 THz while in our case it is significantly above 2.5 THz. This is a key property of optical rectification that enables a spectrally flat dynamic range over a broad range of frequencies as shown in Fig. 2 of the main text which is beneficial for spectroscopy.

The second important distinction relates to pulsed versus continuous-wave terahertz generation. As the referee noted, the HHI group has indeed accomplished a remarkable 132 dB dynamic range in CW [21]. While this specific paper [21] focuses on fiber-coupled PCAs and lacks photonic integration, efforts are underway to realize integrated terahertz emitters on InP in [22] and [18]. To our knowledge, pulsed terahertz generation from integrated PCAs (e.g. on indium selenide platform) is far more challenging [23]. The reason for it lies potentially in the high two-photon absorption of these platforms [24], which we discuss in more detail in the reply to **comment #5**.

To give the community an estimate of the performance and how it compares to PCAs, we do a systematic benchmarking with the PCA under the same conditions.

We compare to "bPCA-180-05-10-1550-h" from Batop GmbH. Since the PCA has a silicon lens on the back side, we attached a silicon lens to the back side of our chip. To ensure a fair comparison, both were measured under the exact same conditions illustrated in Fig. **A3** a. Since we are targeting integrated devices, we chose this commercial device that requires a low bias voltage of 10 V and contains a single antenna, similar to our lithium niobate chip. We compare our device from Fig. 2 to the PCA under two measurement conditions: at 12 mW optical pump power to prevent saturation of the PCA, and at 60 mW where we achieve spectral broadening inside the fiber. As shown in Fig. **A3** d (left panel), at low optical powers (corresponding to pump pulses of 222) the PCA performs better at frequencies below 1 THz (by a factor of approx. 20 dB in power). At frequencies above 1 THz, the two are approximately similar with the TFLN chip being better around 2 THz. We note that the PCA spectrum

we measure corresponds to the one reported in the datasheet [20]. At 60 mW optical power (corresponding to the optical power used in Fig. 2 of the main manuscript), our device exceeds the PCA at frequencies larger than 500 GHz by 10-20 dB. We note that we mounted the Si-lens on our chip manually which may lead to non-ideal alignment. For both measurements, we use 1 MHz modulation frequency, 100 ms integration time, and 25 fs step size.

Despite these efforts, we note that the two devices are quite different, since the PCA does not contain any integrated photonic circuit, limiting miniaturization.

That being said, we agree that it's important to clearly discuss the differences between PCA emission and TFLN emission in our main text and supplementary material. All these works are now referenced in the introduction.

Changes to the manuscript: To address this important remark, we made the following changes:

- (a) we complemented our discussion of bulk terahertz generation in the introduction with an extended discussion on integrated semiconductor technologies, enlarging the scope of our state of the art and underlining the advantages and disadvantages of each platform in lines 53-77
 - (b) we add the supplementary Sec. 6B "Benchmarking of our TFLN emitters to commercial photoconductive antenna"
 - (c) we compare key metrics and properties of various integrated hybrid terahertz devices in supplementary Sec. 6D. "Overview of key metrics and properties for integrated hybrid terahertz devices"
4. What can be done with these devices that cannot be done with the established PCA/diode emitters?

We thank the referee for their intriguing question. Since our focus is on integrated photonic circuits, we answer this question from the perspective of using waveguides in thin film lithium niobate circuits versus waveguides from III-V materials e.g. based on indium phosphide. Perhaps the most fundamental limitation of this semiconductor technology is the small bandgap compared to lithium niobate. This property not only limits the available frequency range for optical pumps (in this case to the near-infrared, excluding for example the visible) but two-photon absorption is knowingly detrimental and limits the highest power values of fs lasers on chip to few mW for Silicon and microwatt for InP and GaAs. This effect is captured by the following simple formula [5]

$$P_{\text{out}} = \frac{P_{\text{in}}}{1 + P_{\text{in}} \cdot \frac{\beta_{\text{TFLN}} L}{A_{\text{eff}} \tau_{\text{rep}}}} \quad (\text{A4})$$

where P_{in} and P_{out} are the average power values at the input and output of the waveguide, L is the propagation length, A_{eff} is the mode area, τ is the pulse length, β_{TPA} is the two-photon absorption coefficient and f_{rep} is the repetition rate. The TPA coefficients for Si, InP, and GaAs are 0.214 cm/GW [5], 24 cm/GW [6], and 2.5 cm/GW [7] accordingly.

To give quantitative numbers for how detrimental this can be, we compute the effect of TPA for a waveguide with dimensions of $A_{\text{eff}} = 0.45\mu\text{m} \times 0.22\mu\text{m}$, $\tau = 60\text{fs}$ and $f_{\text{rep}} = 100\text{MHz}$. These dimensions are typical for example for silicon. As shown in Fig. A4 nonlinear absorption occurs already at microwatt power values, capping the maximal on-chip average power to microwatts. To convince ourselves that these numbers apply in practice, we prepared for this reply to reviewer’s comments silicon waveguides in our cleanroom (since we do not have InP waveguides) and measured the transmitted power after the chip as a function of input optical pump power and fit it to theory A5. We see that the optical power on-chip is limited to a few 100s of microwatts for a waveguide cross-section is $A_{\text{eff}} = 0.45 \mu\text{m}$ by $0.22 \mu\text{m}$, a pump laser with $\tau = 150 \text{ fs}$ and $f_{\text{rep}} = 80 \text{ MHz}$.

Figure A4: Effect of two-photon absorption (TPA) for different semiconductor materials using equation A4 which is derived from [5] We choose a waveguide with dimensions $0.45 \mu\text{m} \times 0.22 \mu\text{m}$ and optical pump $f_{\text{rep}} = 100 \text{ MHz}$, $\tau = 60 \text{ fs}$. The TPA coefficients for Si, InP, and GaAs are 0.214 cm/GW [5], 24 cm/GW [6], and 2.5 cm/GW [7] accordingly. Due to the large bandgap of lithium niobate, power scales linearly in this power range.

This could be the reason for the general absence of terahertz publications with fs pulse propagation in passive photonic integrated InP and GaAs waveguides and the low power values used in silicon photonics [9]. Spintronics also suffer from low damage threshold due to thermal effects [25].

Figure A5: Measured optical output power in a Silicon waveguide under two-photon absorption after 1 mm. Waveguide cross-section is $A_{\text{eff}} = 0.45 \mu\text{m} \times 0.22 \mu\text{m}$. The pump laser has $\tau = 150 \text{ fs}$ and $f_{\text{rep}} = 80 \text{ MHz}$.

In stark contrast, TFLN has been reported to withstand 360 mW average power with 200 fs pulse duration and 18 GHz repetition rate without any damage [4]. This leaves TFLN as the currently most mature platform for femtosecond pulses where both terahertz and optical pulses can be guided on chip.

Finally we wish to point out that TFLN does not require any magnetic biasing (required for terahertz spintronic emitters) that may be difficult to implement, and supports large-area circuits, including for on-chip terahertz applications [8].

Changes to the manuscript: We added the supplementary Sec. 6A "Power handling of common terahertz materials" to highlight the fundamental limit in semiconductors.

5. Is there a way to integrate a detector in a similar way?
<https://doi.org/10.1364/OE.454447>
<https://doi.org/10.1038/s41598-023-40634-3>

We thank the referee for their remark on detection. Terahertz detection in thin film lithium niobate is indeed possible on the same substrate (x-cut TFLN). In the case of generation, two frequency components of the optical pump mix down to the THz domain. In detection, one optical and one THz frequency mix to generate an optical frequency component. Therefore,

both generation and detection can be realized using the same material nonlinearity. We got inspired by the referee’s question and realized a Mach-Zehnder interferometer on chip using our transmission line scheme, where the phase of the optical probe is modulated by the terahertz field in on arm (Fig. **A6** a). Next, phase modulation is transformed to intensity modulation by interference at the output. This intensity change can be then easily measured using a photodiode. By running a single scan with 100 ms integration time at each delay line step, we achieve a modulation of the probe reaching up-to 1.3 % at the terahertz peak (Fig. **A6** b). The Fourier transform show that we can resolve up-to 3 THz and a dynamic range that amounts to 61 dB (Fig. **A6** c). Since we are using a commercial LT-GaAs PCA (iPCA-21-05-1000-800-h from Batop GmbH), the emission drops with THz frequency, in contrast to using our devices for generation or conventional optical rectification in bulk crystals. Similar to generation, we provide the theoretical derivation of terahertz detection which reveals the essential role of our phase-matched transmission lines for achieving this performance.

Co-authors of this manuscript show that that the low optical loss of TFLN waveguides allow realizing quasi-phase matching via antenna arrays, and reconstruct the THz beam profile by using antennas as pixels [8].

To verify the advantage of the transmission line scheme, we measure four devices with varying l_{TL} (Fig. **A7** a). Our results show that we gain in modulation strength (Fig. **A7** b) and dynamic range (Fig. **A7** c) when a longer transmission line is used.

Changes to the manuscript: To address this important remark, we made the following changes:

- (a) We added the section "Broadband terahertz detection" to main text
- (b) We added the theoretical derivation to supplementary Sec 4.A "Theory of terahertz detection inside photonics-integrated transmission lines"
- (c) We added the supplementary Sec. 1.D "Experimental setup for terahertz detection"
- (d) We added the supplementary Sec. 5.B "Effect of transmission line length on terahertz detection"

6. Figure 1. A) It is not clear what can be seen on the picture of the fabricated photonics-integrated terahertz transmission lines.

We thank the referee for pointing out this difficulty.

Changes to the manuscript: We added SEM pictures to Fig. 1 a to visualize the transmission line and optical waveguide.

7. Also, there are many examples of "photonic" structures in the terahertz range, such as photonic crystals, dielectric waveguides, etc.

We thank the referee for their remark. There is indeed a great body of work on photonic THz waveguides which we now reference in the introduction.

Changes to the manuscript: We have expanded significantly our discussion on these available components in the state of the art part of the introduction, lines 80-87.

8. Figure 1. C) Why $n(f_{\text{THz}})$ and not n_{TL} ? This needs to be consistent throughout the manuscript. We thank the referee for their valid point

Changes to the manuscript: $n(f_{\text{THz}})$ was changed to n_{TL} in Fig. 1 c.

9. Fig 1. D) It would be interesting to see the full scale of this graph. We thank the referee for their valid point **Changes to the manuscript:** The full scale has been made visible.

10. Line 79: lowest frequency is missing We thank the referee for noticing the missing value.

Changes to the manuscript: The value 100 GHz has been added to the text in line 123.

11. Is it possible to integrate an optical source with the emitter?

We thank the referee for their question. Indeed, thin film lithium niobate is progressing rapidly, due to the tremendous interest of the community in integrated femtosecond sources. We have mentioned briefly few sources in the conclusion: "These, combined with the mature wafer-scale fabrication of TFLN [26] allows for integration of numerous photonic components such as femtosecond pulse sources [27, 28]".

Changes to the manuscript: No action taken.

12. Line 116: Please state the bandwidth of the optical laser also as a frequency and not only as 75 nm.

We thank the referee for the great remark.

Changes to the manuscript: We added the bandwidth value which corresponds to 9.2 THz in line 165.

13. Why do you distinguish between optical wave and terahertz fields in line 123?

We thank the referee for the valid point.

Changes to the manuscript: The term field has been replaced by wave to avoid confusion in line 171 of the main text.

14. What does the "*" in eq. 3 mean for g_{THz} ? Complex conjugated? But these are not complex functions in the time domain, are they?

We thank the referee for the valid question. The sign "*" means the complex conjugate for g_{THz} . Functions describing optical and terahertz mode considered in this work are real-valued, or, in other words, $g_{\text{THz}} = g_{\text{THz}}^*$. Such notation makes sense only if a global phase is included in the mode profile. To avoid confusion, we decided to make S_{THz} real and remove the complex conjugate

Changes to the manuscript: We have removed the complex conjugate from equation 4

15. Figure caption 2: Is the length of the transmission line really 32 mm? I think this is a typo.

We thank the referee for the valid question. The value is indeed 2 mm.

Changes to the manuscript: The value in the caption has been changed to 2 mm.

16. Lines 162-164: What window function was used to compute the FFT? And please add some details about the data acquisition: Step mode, integration time, and total acquisition time. Especially the acquisition time is important to contextualize the obtained DR.

We thank the referee for the valid questions. We added the details on the time trace. We did not use any window function for the FFT.

Changes to the manuscript: To address this comment, we added in line 214 of the main text "The measurement represents a single time trace, acquired with a step size of 25 fs and an integration time of 100 ms per time point."

17. Line 183: Pure amplitude modulation is of course not a terahertz communication will be realized. There are better modulation schemes.

We thank the referee for their valid point. Our demonstration of amplitude modulation is indeed a very initial step for potentially realizing more complex communication schemes. One could envision ways to achieve also phase modulation, for example by multiplexing and poling techniques, but these require more in-depth studies.

Changes to the manuscript: We changed the introduction of this section in lines 238-240 as follows "Amplitude modulation is one of the cornerstones of current communication schemes and precise control over the amplitude of the generated terahertz fields is essential for future generations of terahertz devices".

18. Line 196: Why is the 120 μm trace not shown anywhere? Only 2 mm, 100 μm and 200 μm . Why don't you show traces for all lengths?

We thank the referee for the question. We are unsure if we understood the question correctly but the 120 μm trace is shown in Fig. 3b in the main

text. We have shown the measurements for three devices with identical antennas and different transmission line length in Supplementary Fig. 20. For the cases of 100 μm and 200 μm , the cavity effect both the time and frequency domain results. Therefore, they are shown separately in Fig. 4.

Changes to the manuscript: no action taken.

19. Lines 219-220: Two times odd modes for strong and weak excitation. One is wrong.

We thank the referee for noting this mistake. According to our definition of the modes, the even modes are trapped in the cavity.

Changes to the manuscript: Sentence in Lines 275-277 was changed to "Therefore, an antenna placed at the center can extract the odd modes into the far field (strong extraction), while not efficiently out-coupling even modes into the far-field (weak extraction)".

20. Figure 4. C) The 200 μm transmission line shows a double pulse with the same amplitude. What I would expect from a transmission line filter is a pulse train with increasing amplitude (depending on the Q-factor).

We thank the referee for their valid question. The amplitude of the pulse train is not increasing because the pump propagates along the device only once (single pass). After the optical pump leaves the strip-line cavity, the terahertz radiation does not experience any parametric gain anymore, which prevents a terahertz cavity build-up. When the terahertz is reflected at the end of the transmission line, it travels backwards and then forward again, after being reflected a second time. In this way, the cavity formed by the transmission line which shapes the spectrum of the emitted terahertz radiation. Because of the terahertz loss (described in details in Supplementary Sec. 2C) the amplitude of the reflected fields continuously decreases. What the referee suggested can be achieved by using a multi-pass configuration where an optical pump would repetitively pump the terahertz cavity. This could be achieved with lasers with GHz repetition rate values and matching the cavity length to the repetition rate (e.g. 5.4 mm cavity for 25 GHz repetition rate). Such scheme could work for sub-THz frequencies since the absorption is relatively low as shown in Supplementary Fig. 8, but needs extensive studies.

Changes to the manuscript: to clarify the cavity mechanism, we write the following "The remaining terahertz field continues to build up until it reaches the end of the transmission line and reflects back. The optical pump leaves the structure, and the remaining terahertz radiation experiences net loss during propagation. Once the terahertz pulse reaches the antenna again, it is coupled out with a group delay of $t_g = \frac{n_g L_{TL}}{c}$ with respect to the initial terahertz pulse."

21. Why is your Q-factor so low? There must be a huge impedance mismatch at both ends of the line. Are the losses underestimated?

We thank the referee for the question. As shown in the simulation results in Fig. 4 b, the reflection is sufficiently high for strong cavity effects but the quality factor is limited by the propagation losses of the terahertz waves. We have created a model and used our simulated loss values in Supplementary Sec. 2D. We compare the simulation to the measurement results in Supplementary Fig. 13 and obtain good agreement of the temporal decay of the terahertz pulses and the spectral features. We have modified the final text in this section to clarify the origin of this limitation to the readers.

Changes to the manuscript: The description of the results have been modified as follows: "Our full cavity model shows that the quality factor is limited by the losses inside the terahertz transmission line. The model captures faithfully the experimental details such as field amplitudes in the time domain, spacing of the measured resonances in the frequency domain, and their linewidth (full details in Supplementary Information Sec. 2.D)"

22. Line 280: The term "terahertz speed modulators" seems a bit strange to me. Maybe it is better to call them "modulators operating at terahertz frequencies".

We thank the referee for the valid point. We changed the wording as recommended.

Changes to the manuscript: We rewrite the sentence to "This is beneficial in the context of modulators operating at terahertz frequencies".

23. Line 422: The waveguides were patterned using electron beam lithography: Does the platform also allow conventional lithography?

We thank the referee for the excellent question. Yes, absolutely. Deep UV lithography has been demonstrated with low roughness values leading to 2 dB/cm losses [29]. However, most of the current work is based on e-beam lithography due to its higher resolution. In our case, this is important for example to place the transmission line around the optical waveguide with sub-100 nm accuracy.

Changes to the manuscript: no action was taken.

24. Why is the 200-fs probe pulse so long? Usually you want to use the shorter pulse as the probe pulse.

We thank the referee for the valid point and apologize for the typo. The pulse length is indeed 58 fs and not 200 fs.

Changes to the manuscript: We write the following text "The pulse length of the 780 nm probe is 58 fs."

25. Why is the refractive index measured at about 1500 nm and not at 1550 nm?

We thank the referee for the valid point. We replaced the data with one measured at 1550 nm. The group index at 1550 nm is 2.25 instead of 2.24

at 1500 nm. Hence our reasoning and phase matching criteria are still valid.

Changes to the manuscript: Figure and measurement have been changed in the Supplementary Sec. 1B

26. **Supplement Figure 5 c):** Black and brown are not good choices for coloring. We thank the referee for their comment.

Changes to the manuscript: Supplementary Fig. 5 has been changed.

References

- [1] K. Yoshioka, G. Bernard, T. Wakamura, M. Hashisaka, K.-i. Sasaki, S. Sasaki, K. Watanabe, T. Taniguchi, and N. Kumada. “On-chip transfer of ultrashort graphene plasmon wave packets using terahertz electronics.” *Nature Electronics*, **7**(7):537–544 (2024).
- [2] A. M. Potts, A. K. Nayak, M. Nagel, K. Kaj, B. Stamenic, D. D. John, R. D. Averitt, and A. F. Young. “On-chip time-domain terahertz spectroscopy of superconducting films below the diffraction limit.” *Nano Letters*, **23**(9):3835–3841 (2023).
- [3] A. Herter, A. Shams-Ansari, F. F. Settembrini, H. K. Warner, J. Faist, M. Lončar, and I.-C. Benea-Chelmus. “Terahertz waveform synthesis in integrated thin-film lithium niobate platform.” *Nature Communications*, **14**(1):11 (2023).
- [4] M. Ludwig, F. Ayhan, T. M. Schmidt, T. Wildi, T. Voumard, R. Blum, Z. Ye, F. Lei, F. Wildi, F. Pepe, et al. “Ultraviolet astronomical spectrograph calibration with laser frequency combs from nanophotonic lithium niobate waveguides.” *Nature Communications*, **15**(1):7614 (2024).
- [5] X. Sang, E.-K. Tien, and O. Boyraz. “Applications of two photon absorption in silicon.” *Journal of optoelectronics and advanced materials*, **11**(1):15 (2009).
- [6] D. Vignaud, J.-F. Lampin, and F. Molot. “Two-photon absorption in InP substrates in the 1.55 μm range.” *Applied physics letters*, **85**(2):239–241 (2004).
- [7] W. C. Hurlbut, Y.-S. Lee, K. Vodopyanov, P. Kuo, and M. Fejer. “Multi-photon absorption and nonlinear refraction of GaAs in the mid-infrared.” *Optics Letters*, **32**(6):668–670 (2007).
- [8] A. Tomasino, A. Shams-Ansari, M. Lončar, and I.-C. Benea-Chelmus. “Large-area photonic circuits for terahertz detection and beam profiling.” arXiv preprint arXiv:2410.20407 (2024).

- [9] I.-C. Benea-Chelmus, Y. Salamin, F. F. Settembrini, Y. Fedoryshyn, W. Heni, D. L. Elder, L. R. Dalton, J. Leuthold, and J. Faist. “Electro-optic interface for ultrasensitive intracavity electric field measurements at microwave and terahertz frequencies.” *Optica*, **7**(5):498–505 (2020).
- [10] L. Smith, V. Shiran, W. Gomaa, and T. Darcie. “Characterization of a splitting-resonator-loaded transmission line at terahertz frequencies.” *Optics Express*, **29**(15):23282–23289 (2021).
- [11] D. Zhu, L. Shao, M. Yu, R. Cheng, B. Desiatov, C. Xin, Y. Hu, J. Holzgrafe, S. Ghosh, A. Shams-Ansari, et al. “Integrated photonics on thin-film lithium niobate.” *Advances in Optics and Photonics*, **13**(2):242–352 (2021).
- [12] S. Shekhar, W. Bogaerts, L. Chrostowski, J. E. Bowers, M. Hochberg, R. Soref, and B. J. Shastri. “Roadmapping the next generation of silicon photonics.” *Nature Communications*, **15**(1):751 (2024).
- [13] M. Theurer, T. Göbel, D. Stanze, U. Troppenz, F. Soares, N. Grote, and M. Schell. “Photonic-integrated circuit for continuous-wave THz generation.” *Optics Letters*, **38**(19):3724–3726 (2013).
- [14] S. Jia, M.-C. Lo, L. Zhang, O. Ozolins, A. Udalcovs, D. Kong, X. Pang, R. Guzman, X. Yu, S. Xiao, et al. “Integrated dual-laser photonic chip for high-purity carrier generation enabling ultrafast terahertz wireless communications.” *Nature communications*, **13**(1):1388 (2022).
- [15] L. Schwenson, L. Liebermeister, F. Walter, S. Nellen, M. Schell, and R. B. Kohlhaas. “Photonic Integrated Continuous Wave Terahertz Spectrometer with 90 dB Dynamic Range and 4 THz Bandwidth.” In “2024 49th International Conference on Infrared, Millimeter, and Terahertz Waves (IRMMW-THz),” pages 1–2. IEEE (2024).
- [16] M. Nickerson, B. Song, J. Brookhyser, G. Erwin, J. Kleinert, and J. Klamkin. “Gallium arsenide optical phased array photonic integrated circuit.” *Optics Express*, **31**(17):27106–27122 (2023).
- [17] P. A. Verrinder, L. Wang, J. Fridlander, F. Sang, V. Rosborough, M. Nickerson, G. Yang, M. Stephen, L. Coldren, and J. Klamkin. “Gallium arsenide photonic integrated circuit platform for tunable laser applications.” *IEEE Journal of Selected Topics in Quantum Electronics*, **28**(1: Semiconductor Lasers):1–9 (2021).
- [18] M. Deumer, S. Nellen, S. Breuer, R. B. Kohlhaas, L. Schwenson, K. Wenzel, L. Liebermeister, M. Schell, and B. Globisch. “Waveguide-integrated photoconductive THz receivers.” In “2022 47th International Conference on Infrared, Millimeter and Terahertz Waves (IRMMW-THz),” pages 1–2. IEEE (2022).

- [19] S. Nellen, T. Ishibashi, A. Deninger, R. Kohlhaas, L. Liebermeister, M. Schell, and B. Globisch. “Experimental comparison of UTC-and PIN-photodiodes for continuous-wave terahertz generation.” *Journal of Infrared, Millimeter, and Terahertz Waves*, **41**:343–354 (2020).
- [20] “datasheet.” https://batop.de/products/terahertz/photoconductive-antenna/data-sheet/1550nm/data_sheet_PCA-180-05-10-1550.pdf.
- [21] M. Deumer, S. Breuer, S. Berrios, S. Keyvaninia, G. Schwanke, L. Schwen-son, S. Lauck, L. Liebermeister, S. Nellen, M. Schell, et al. “Continuous wave THz receivers with rhodium-doped InGaAs enabling 132 dB dynamic range.” *Optics Express*, **32**(17):29855–29867 (2024).
- [22] S. Nellen, S. Lauck, E. Peytavit, P. Szriftgiser, M. Schell, G. Ducournau, and B. Globisch. “Coherent wireless link at 300 GHz with 160 Gbit/s enabled by a photonic transmitter.” *Journal of Lightwave Technology*, **40**(13):4178–4185 (2022).
- [23] P. Chen, M. Hosseini, and A. Babakhani. “An integrated germanium-based THz impulse radiator with an optical waveguide coupled photoconductive switch in silicon.” *Micromachines*, **10**(6):367 (2019).
- [24] E. Bente, S. Andreou, Y. Jiao, and K. Williams. “Effects of two-photon absorption and non-linear index in inp-based passive waveguides on integrated extended cavity semiconductor lasers.” In “2021 Conference on Lasers and Electro-Optics Europe & European Quantum Electronics Conference (CLEO/Europe-EQEC),” pages 1–1. IEEE (2021).
- [25] F. Paries, F. Selz, C. N. Santos, J.-F. Lampin, P. Koleják, G. Lezier, D. Troadec, N. Tiercelin, M. Vanwolleghem, A. Addda, et al. “Optical damage thresholds of single-mode fiber-tip spintronic terahertz emitters.” *Optics Express*, **32**(14):24826–24838 (2024).
- [26] K. Luke, P. Kharel, C. Reimer, L. He, M. Loncar, and M. Zhang. “Wafer-scale low-loss lithium niobate photonic integrated circuits.” *Opt. Express*, **28**(17):24452–24458 (2020).
- [27] M. Yu, D. Barton III, R. Cheng, C. Reimer, P. Kharel, L. He, L. Shao, D. Zhu, Y. Hu, H. R. Grant, et al. “Integrated femtosecond pulse generator on thin-film lithium niobate.” *Nature*, **612**(7939):252–258 (2022).
- [28] X. Wang, Z. Li, J. Chen, C. Shang, Z. Zhang, H. Li, Y. Liu, C. Zeng, and J. Xia. “Integrated thin-film lithium niobate electro-optic frequency comb for picosecond optical pulse train generation.” *Applied Physics Letters*, **124**(20):201101 (2024). eprint: <https://pubs.aip.org/aip/apl/article-pdf/doi/10.1063/5.0206281/19945299/201101.1.5.0206281.pdf>.
- [29] M. Zhang and K. Luke. “Lithium niobate devices fabricated using deep ultraviolet radiation.” (2021). US Patent 11,086,048.

Figure A6: Photonics-integrated broadband terahertz detector. **a** Optical setup used for the characterization of the terahertz detector. The detected terahertz radiation is emitted from a commercial GaAs-based photoconductive antenna pumped at 780 nm. The probe at 1560 nm is coupled to a highly nonlinear fiber (HNLF) to compress the pulse length down to 60 fs. Broadband terahertz detection in TFLN is implemented through on-chip electro-optic sampling. In electro-optic sampling, a terahertz electric field modulates the phase of a femtosecond probe. Measuring this phase allows reconstructing the incident terahertz signal. In our implementation, the probe pulse is split in power to propagate along two arms of a Mach-Zehnder interferometer. In one arm, a dipole antenna captures the terahertz radiation which propagates inside the transmission line to provide phase modulation to the probe pulse via the electro-optic (Pockels) effect. This phase-modulated probe propagates further to the output splitter of the interferometer, where it interferes with the probe that propagated along the second arm. This interference results in an intensity modulation at the output of the interferometer that is measured with a photodiode. **b** The upper panel shows the terahertz waveform retrieved through electro-optic sampling in a bulk gallium phosphide (GaP) crystal which is 200 μm thick and has a flat frequency response. This allows accurate reconstruction of the input terahertz pulse. The middle panel shows the normalized modulation of the probe power from our TFLN device with the length $L_{\text{ant}} = 200 \mu\text{m}$ and a transmission line length $l_{\text{TL}} = 1 \text{ mm}$. The lower panel shows the terahertz signal as-detected with a commercial photoconductive antenna (PCA). **c** Fourier transforms of the signals in **b** compares the various detection schemes, showcasing the ability of our TFLN chip to reach 3 THz bandwidth with 60 dB dynamic range.

Figure A7: **Dependence of THz detection on the length of the transmission line l_{TL} .** **a** terahertz detection from four devices with the same antenna length $l_{ant} = 200 \mu\text{m}$ and varying transmission line length $l_{TL} = 125 \mu\text{m}, 250 \mu\text{m}, 500 \mu\text{m}, 1 \text{ mm}$. For emission, we pump a commercial LT-GaAs at 780 nm (iPCA-21-05-1000-800-h from Batop GmbH). **b** extracted peak modulation values of the time domain traces in percentage. **c** spectra of the THz signal. Dashed lines highlight the roll-off with frequency.

Response to referees

We thank the referees for assessing our revised manuscript. We address the remaining comments below. The referee's comments are shown in blue and our reply is shown in black. Changes to the manuscript are shown in red.

Reviewer #1

1. The additional analysis and revisions have indeed improved the manuscript. As the second referee also noted, the main competition to the demonstrated THz source and detector are photoconductive antennas (PCAs). Numerous prior studies have demonstrated THz signal generation and detection using on-chip PCA sources and detectors integrated with THz transmission lines (as referenced in my previous comments). These on-chip PCAs achieve significantly higher terahertz powers (on the order of milliwatts) and higher SNRs compared to the TFLN devices presented in this manuscript, which are expected to generate only 0.2 nW of power. Unfortunately, the authors dismiss PCAs by stating that they require a free-space optical pump, while their TFLN devices also rely on a free-space optical pump. The only difference is that the free-space pump is coupled into an optical waveguide in the TFLN devices to satisfy a phase-matching condition, a requirement that does not apply to PCAs. Therefore, the manuscript does not convincingly demonstrate how the TFLN devices advance the current state of the art.

We thank the referee for the summary and for highlighting the improvements from the previous submission.

We also thank the referee for raising the second point. The referee is right, PCAs are a great technology that goes back 40 years to the very beginnings of time-domain measurements on picosecond timescales. Seminal papers on electro-optic sampling by Vladmanis, Mourou and Gabel (1983) and Mourou and Mayer (1984) exploited the photoconductive effect and transmission lines. Since then, many works make use of transmission lines for terahertz delivery, including the two references the referee mentions, see below:

- Yoshioka, et al., 2024. On-chip transfer of ultrashort graphene plasmon wave packets using terahertz electronics. *Nature Electronics*, pp.1-8.

- Potts, et al., 2023. On-chip time-domain terahertz spectroscopy of superconducting films below the diffraction limit. *Nano Letters*, 23(9), pp.3835-3841.

On the other hand, thin film lithium niobate platform has attracted great attention from many groups and companies, and the focus of our work is to bring terahertz capabilities to this highly promising platform. We believe that this platform will play a major role in the future due to the various advantages we state in the conclusion and outlook.

All experimental data we show regarding

1. Generation, Fig. 2
2. EO control, Fig. 3
3. integrating terahertz transmission line cavities with optical waveguides as shown in our Fig. 4,
4. broadband terahertz detection through electro-optic sampling, as shown in Fig. 5. We note that electro-optic sampling has emerged as an extremely powerful technique for quantum sensing, as discussed in the recent *Optica* Review paper Ileana-Cristina Benea-Chelmus, Jérôme Faist, Alfred Leitenstorfer, Andrey S. Moskalenko, Ioachim Pupezza, Denis V. Seletskiy, and Konstantin L. Vodopyanov, "Electro-optic sampling of classical and quantum light," *Optica* **12**, 546-563 (2025)

is state-of-the-art in thin film lithium niobate terahertz photonics. We believe that adding these terahertz functionalities to thin film lithium niobate may pave the way for entirely new applications as we envision in lines 137-141, 406-411, 423-431, and 435-447.

The way we see it, is that in the end, both platforms will co-exist, each one for its own merits. However, we fully agree that higher SNR and power values are desired, and these will be subject of follow-up studies. We see several improvements that could lead to significantly higher average power and SNR due to the nonlinear dependency of conversion efficiency on various parameters such pulse length, power on chip and coupling efficiency (lines 382-384), and transmission line lengths (224-227). The improvements are quantified in the above-mentioned lines.

Changes to the manuscript: To address this comment from the referee, we took the following steps:

- We understand that the terms “integrated” and “on-chip” may easily be misunderstood. To avoid this, we now make the distinction between optically “top-illuminated” photoconductive emitters/receivers and optical waveguide-coupled photoconductive emitters.
- We note that a new paper appeared in *APL photonics* 10 (3), 2025 while our paper was with the referees, which demonstrates precisely the advantages of waveguide-coupled terahertz photoconductive receivers compared to top-illuminated ones, achieving 22-fold increase in photoresponse, a 500-fold improvement in THz responsivity, and a 4.7-fold reduction in noise-equivalent power compared to state-of-the-art top-illuminated PCAs. We now also added to the manuscript and supplementary material. We reference it accordingly in lines 57-60.
- To avoid misunderstandings, we now state the amount of average thz power achieved in pulsed of ref. [33] in the text.
- To avoid confusion, we have edited the sentence "We empathize however that our chip is an integrated platform while the PCA top-illuminated" in Supplementary Sec. 6B

Reviewer #2

1. I would like to thank the authors for their detailed response. All of my critical comments have been addressed thoroughly and appropriately. I therefore recommend accepting the manuscript in its current form. I do have one minor remark: As the authors mention, some of the best PCAs come from HHI. A comparison with these antennas would therefore be worthwhile. I am, of course, aware that HHI technology has been developed and refined over many years. Nevertheless, Batop antennas should not be considered a reference. However, since the focus here is more on replacing power measurements, the approach taken is acceptable

We thank the referee for the positive assessment of our revision and recommending its publication in its current state.